**Data Availability Statement:** All relevant data are within the manuscript and its Supporting Information files.

# Microdose lithium improves behavioral deficits and modulates molecular mechanisms of memory formation in female SAMP-8, a mouse model of accelerated aging

Arthur Antonio Ruiz Pereira[ID][1,2], Alessandra Macedo Pinto[3], Helena Nascimento Malerba[1,2], Mariana Toricelli[2], Hudson Sousa Buck[2,4], Tania Araujo Viel[ID][1,2,3]*

1 Department of Pharmacology, Institute of Biomedical Sciences, Graduate Course on Pharmacology, Universidade de Sao Paulo, Sao Paulo, Sao Paulo, Brazil, 2 Laboratory of Neuropharmacology of Aging, School of Arts, Sciences and Humanities, Universidade de São Paulo, Sao Paulo, Sao Paulo, Brazil, 3 Graduate Course on Gerontology, School of Arts, Sciences and Humanities, Universidade de São Paulo, Sao Paulo, Sao Paulo, Brazil, 4 Department of Physiology, University of Mogi das Cruzes, Mogi das Cruzes, Sao Paulo, Brazil

* taniaviel@usp.br

## Abstract

Alzheimer's disease (AD) is the most common neuronal disorder that leads to the development of dementia. Until nowadays, some therapies may alleviate the symptoms, but there is no pharmacological treatment. Microdosing lithium has been used to modify the pathological characteristics of the disease, with effects in both experimental and clinical conditions. The present work aimed to analyze the effects of this treatment on spatial memory, anxiety, and molecular mechanisms related to long-term memory formation during the aging process of a mouse model of accelerated aging (SAMP-8). Female SAMP-8 showed learning and memory impairments together with disruption of memory mechanisms, neuronal loss, and increased density of senile plaques compared to their natural control strain, the senescence-accelerated mouse resistant (SAMR-1). Chronic treatment with lithium promoted memory maintenance, reduction in anxiety, and maintenance of proteins related to memory formation and neuronal density. The density of senile plaques was also reduced. An increase in the density of gamma-aminobutyric acid A (GABA$_A$) and α7 nicotinic cholinergic receptors was also observed and related to neuroprotection and anxiety reduction. In addition, this microdose of lithium inhibited the activation of glycogen synthase kinase-3beta (GSK-3β), the classical mechanism of lithium cell effects, which could contribute to the preservation of the memory mechanism and reduction in senile plaque formation. This work shows that lithium effects in neuroprotection along the aging process are not related to a unique cellular mechanism but produce multiple effects that slowly protect the brain along the aging process.

**Funding:** TAV 2016/07115-6 and 2020/14133-6
MT 2017/21655-6 The Sao Paulo Research
Foundation (FAPESP) www.fapesp.br AARP, AMP,
HNM received student fellowship from:
Coordenação de Aperfeiçoamento de Pessoal de
Nível Superior – Brasil (CAPES) – Finance Code
001 https://www.gov.br/capes/pt-br The sponsors
did not play any role in the present work.

**Competing interests:** The authors have declared
that no competing interests exist.

## Introduction

Memory impairment is still the most relevant and early symptom of Alzheimer's disease among the cognitive domains. As a way to construct experiences and personality, memory is the capacity to acquire, save, and evoke information, which demands morphological changes in synapses [1]. The increase in strengthening and quantity of neuronal connexions is referred to as neuroplasticity and is the most accepted model to explain long-term memory formation. The mechanism for neuroplasticity occurs in the hippocampus and other brain regions due to synaptic changes that last for an extended period after gene transcription, resulting in an increase in the efficacy of synaptic transmission that can also be modulated by epigenetic mechanisms [2,3]. Although the molecular and electrophysiological mechanisms of long-term memory are well known, there is no efficient pharmacological strategy capable of treating or restoring the structural and functional loss that happens in AD and other neurodegenerative diseases.

Lithium was first described to help treat manic episodes by the psychiatrist John Cade in 1949 [4]. Later, it was approved for the treatment of mania and bipolar disorders by the American Agency Food and Drug Administration in 1970. Until today, it is considered an efficient drug for the treatment of manic episodes, depression, and mixed episodes [5]. More recent works show that lithium has a modulator effect in several psychiatric, neurological, and neurodegenerative processes, such as anxiety, Alzheimer's disease, Parkinson's disease, and ischemic stroke, among others [6–10].

Despite the evident benefits of lithium salts to bipolar disorders and other psychiatric and neurological diseases, the main limiting factor for its prescription by psychiatrists is the side effects that this metal produces, as reported recently [5]. In this way, our group and others have used lithium salts in low-dose, and the benefits were evident in clinical trials, mouse models of Alzheimer's disease, and human cells [11–16]. In our previous works, we used the term "microdose lithium" as the lithium dose for humans was 300 μg administered daily (1.5 mg of lithium carbonate—[15]) and the dose of 5 μg/day/animal (0,25 mg/kg/day) for mice [14]. Many groups have shown that low doses of Lithium (15–300 μg) are efficient against dementia and have little side effects in long-term protocols [10,17–19]. However, these doses are not in current usage by physicians to prevent Alzheimer's disease or as an adjuvant strategy to mild cognitive impairment treatment. Our hypothesis for this fact is that there are not enough experimental data with biological mechanisms to support its clinical use. In this way, the aim of the present work was to verify if the chronic use of lithium in low dose could maintain the density of proteins related to memory formation. Long-term spatial memory and anxiety were verified in senescence-accelerated mice prone(SAMP-8), a mouse model of accelerated aging and Alzheimer's disease (Takeda, 1999; Morley et al., 2012). SAMP-8 lives in approximately 50% of normal mice (roughly 12 months). In this period, they present declines in learning and memory processing, emotional alterations (like anxiety and depression-like behaviors), abnormal circadian rhythms, and brain atrophy (Takeda, 1999). SAMR-1 animals, used as controls, have regular aging development. These results obtained in the present work further support the chronic use of this metal in low doses to delay the evolution of the behavioral effects of neurodegenerative processes.

## Materials and methods

### Animals

The accelerated aging mouse line comes from AKR/J animals. In 1968, at Kyoto University, after brother-sister crossings, some litters showed some different characteristics such as

decreased activity, periophthalmic lesions, and early death, but without any evidence of mal-formation. This new phenotype presented was inherited by other generations [20]. Years later, in 1975, the litters that presented the phenotypes described above were isolated as "senescence-prone" parents, and the animals that did not have an altered life course were "senescence-resis-tant" parents [20].

Nowadays there are several lineages of SAMP animals; the isolation of these lineages occurred due to the different characteristics they presented. SAMP-8 animals present the hall-marks of accelerated aging, early death, memory and learning deficits, presence of amyloid pla-ques in the hippocampus, blood-brain barrier dysfunction, loss of neurons, and cortical atrophy, as described by other groups [21,22].

Two breeding families of senescence-accelerated mice prone (SAMP-8) and senescence-accelerated mice resistant (SAMR-1) were kindly donated by Dr. Eliana Akamine (ICB-USP), and a colony was established in the animal facility of the Department of Physiological Sciences of Santa Casa de Sao Paulo School of Medical Sciences, Sao Paulo, Brazil. 24 female SAMP-8 and 11 female SAMR-1 were maintained in groups of 2–6 in ventilated cages (Alesco, Brazil) with food and water *ad libitum*. Controlled room temperature (24–26˚C), humidity (55%), and a 12-hour light/dark cycle were maintained in the animal facility, and mice were kept until they completed months of age (the dead-point established in our protocol).

All animal protocols were performed according to the ethics principle for the use of labora-tory animals of the Brazilian Society of Laboratory Animal Science (SBCAL, Brazil) and the Guide for the Care and Use of Laboratory Animals (National Institutes of Health, Publication number 86–23, Bethesda, MD). The proceedings were approved by the Animal Ethics Com-mittee of the Santa Casa de Sao Paulo School of Medical Sciences (protocol number 12/2016), and all effort was made to minimize the number of animals and their suffering, following the 3R's principle (Replacement, Reduction, and Refinement).

## Treatment with microdose lithium

Lithium, in the form of lithium carbonate ($Li_2CO_3$), was dissolved in water and administered *ad libitum*. The dose used was the same as published by our research team before for mice treatment [13,14] and proportional to that used in a previous manuscript with Alzheimer's dis-ease's patients [15]. The dose used in mice corresponds to 1.5 mg of lithium carbonate/day or 0.006 mEq of lithium/kg. The dose was adjusted according to the mice's pharmacokinetics profile [23]. The final dose was 5ug/day/animal, corresponding to 0.25 mg/kg/day.

## Experimental design

For behavioral experiments, it was used 11 female SAMR-1 and 24 female SAMP-8 divided into two groups: untreated (n = 12) and treated with $Li_2CO_3$ *ad libitum* (n = 12).

The timeline of the experimental design is shown in Fig 1.

## Behavioral tests

**Evaluation of spatial memory with Barnes maze.** Spatial memory was evaluated using the Barnes maze, following the protocol of previous work (Baraldi et al., 2013; Balthazar et al., 2018; Malerba et al., 2021). Briefly, each animal was placed in the center of a white, 100 cm diameter board with 30 holes arranged radially. Under one of the holes, a dark box was estab-lished as an escape box. A fluorescent lamp was placed above the center of the arena, and a radio "out of frequency" was played. Animals had 5 min to find the escape box. Once inside the box, the lights and radio were turned off, and the animal was left quiet for 1 minute. The learning phase was performed when animals were two months old, and sessions were repeated

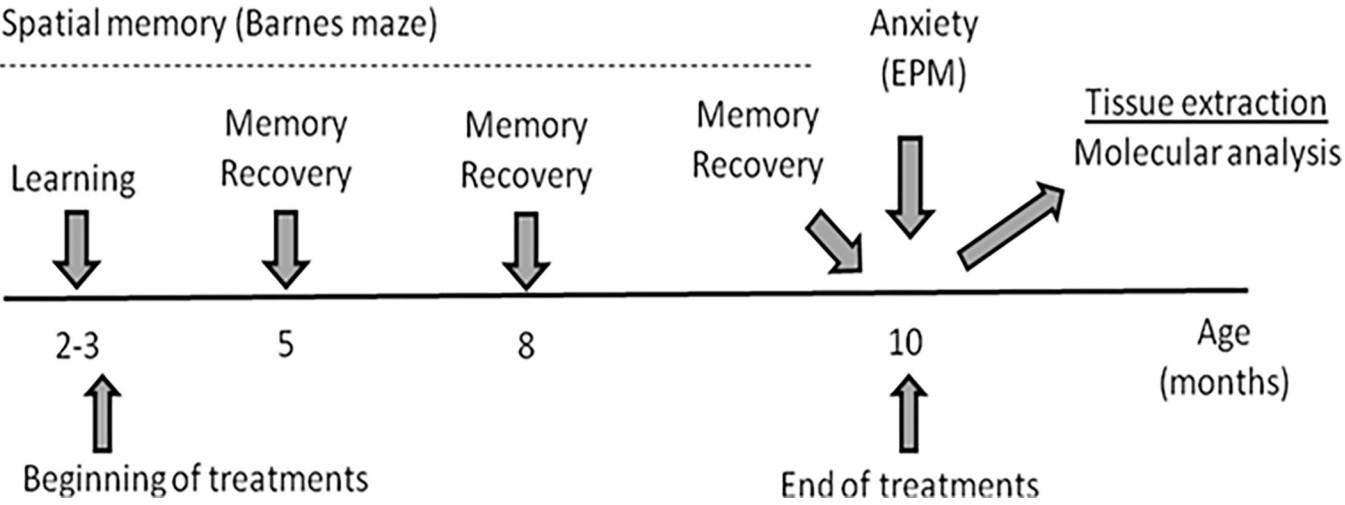

**Fig 1. Timeline for the experimental design.** SAMR-1 and SAMP-8 groups were treated or not with lithium salt *ad libitum* for 7 to 8 months. In the end, animals were submitted to behavioral tests soon after they were anesthetized and killed for tissue extraction.

once a day for five consecutive days (Acquisition Session). The task was repeated when animals were 5, 8, and 10 months old to assess memory recovery. The time to enter the escape box and the number of wrong holes explored before it entered the escape box were registered.

**Evaluation of anxiolytic-like effects with the elevated plus maze.** The Elevated Plus Maze comprised two open opposed arms (50 x 10 cm) crossed by two closed opposed arms (50 x 10 cm and 40 cm high walls). The maze was maintained 50 cm from the floor. The percentage of the number of entries in open arms over total entries in open and closed arms was registered. Also, the percentage of time spent in open arms concerning whole time was calculated. In addition, the number of head-dippings (when the animal dipped the head to explore the environment outside of the maze) was also registered. Animals were submitted to this test when they reached ten months of age.

## Biochemical and histological analysis

After behavioral tests, mice were anesthetized with 4% isoflurane, and the brain and blood were collected and immediately frozen in dimethylbutane at -50˚C and stored at -80˚C until used. One hemisphere was used for western blot and ELISA analysis, whereas the other was used for tissue analysis.

**Analysis of the density of proteins related to neuroplasticity and long-term memory.** Hippocampal tissues were isolated and homogenized in lysis buffer containing 50 mM Tris–HCl (pH 7.4), 0.1% Triton X-100, 4 mM ethylene glycol tetraacetic acid (EGTA), 10 mM ethylenediamine tetraacetic acid (EDTA) and a tablet with protease inhibitors (Complete Protease Inhibitor Cocktail Tablets, Roche, Basileia, Switzerland) and phosphatase inhibitors (Phos-STOP Phosphatase Inhibitor Cocktail, Roche, Basileia, Switzerland). Samples were centrifuged (12,000 rpm, 15 min, 4˚C), the supernatant was separated, and the total protein was determined with the Bradford method (1976). After that, proteins were separated by 10% sodium dodecyl sulfate-polyacrylamide gel electrophoresis (SDS-PAGE) and transferred onto nitrocellulose membranes. To ensure equal protein loading (Salinovich and Montelaro, 1986) and to evaluate the loading quality, the Ponceau S Red (Sigma-Aldrich, St-Louis, Missouri, USA) method was used to stain the membranes before probing with the antibodies.

Membranes were preincubated with Tris buffer saline (Sigma-Aldrich, St-Louis, Missouri, USA) with 0.1% Tween-20 (TBST) and 1% bovine serum albumin (BSA, Sigma-Aldrich, St-Louis, Missouri, USA) for 1h in room temperature. Following that, membranes were incubated overnight at 4˚C with the following primary antibodies, diluted in the same buffer: β-actin (Cell Signaling, 4970, 1:1000); synaptophysin (Abcam, ab8049,1:500); postsynaptic density protein 95 (PSD95, Cell Signaling, 2507, 1:1000—a protein that stabilizes the synaptic expression of glutamate receptors); N-methyl-D-aspartate receptor subunit R2B (NMDA-R2B, Abcam, ab65783, 1:1000—a receptor for the neurotransmitter glutamate, involved in memory formation); α-amino-3-hydroxy-5-methyl-4-isoxazolepropionic acid (AMPA) receptor (GluA2/3/4 subunits, Cell Signaling, 2460, 1:1000 another receptor for glutamate that is also involved in memory formation), gamma aminobutyric acid, subunit A receptor (GABA$_A$, Abcam, ab33299, 1:5000—a receptor for the neurotransmitter GABA, involved in modulation of memory and anxiety, among other functions), α7 nicotinic receptor (Millipore, AB15332, 1:1000—a receptor involved in the control of glutamate release and modulation of memory formation and recovery); calcium/calmodulin-dependent protein kinase IV (CaMK-IV, Cell Signaling, 4032, 1:1000—a protein activated by increases in the concentration of calcium and calmodulin as a consequence of glutamate receptors activation); glycogen synthase kinase-3 beta (GSK-3β, Cell Signaling, 12456, 1:1000—one of the principal pharmacological targets for lithium in the cells); phosphorylated glycogen synthase kinase-3 beta (pGSK-3β (Ser 9), Cell Signaling, 5558P, 1:1000—the increase in phosphorylation of GSK-3β inhibits this protein, leading to cell survival and neuroprotection). The membranes were then washed and incubated with a secondary biotinylated-antibody horseradish peroxidase-conjugated anti-IgG (Millipore, AP187P, 1:4000) for 2 h at room temperature. The immune complexes were detected using chemiluminescence (Western Lightning ECL Pro, Perkin Elmer, NEL120001EA) read in Image Quant Las 500 (GE). The density of immunoblotting was quantified with Image J software 1.8.0 (*National Institutes of Health*, Bethesda, Maryland, EUA).

**Evaluation of neurons and astrocyte densities in the hippocampus.** For immunofluorescence of neuronal and astrocytic labeling, the hemispheres of 5 animals per group were separated, and 20 μm samples were obtained in a cryostat (-22˚C to -20˚C, Microm HM 404N, Francehville, France). Brain sections were placed in gelatin-coated slides and kept at -80˚C until use.

For neuronal labeling, samples were fixed with 100% acetone for 5 min, washed with PBS (pH 7.4) for 1 min, and incubated with 0.6% $H_2O_2$ in methanol for 20 min. After that, slides were washed in PBS for 5 min and incubated with *normal serum* for 1 hour. Following that period, samples were incubated with an anti-NeuN primary antibody (Millipore, ABN78, 1:500) for 1 hour again. After that, samples were washed with PBS for 5 min and incubated with the secondary antibody Alexa Fluor 597 (goat anti-rabbit IgG, Life Technologies, 1:200) for 2 hours. At the end of this incubation, samples were washed with PBS for 5 min and, soon after, incubated with an anti-GFAP antibody (Abcam, ab16997, 1:200) for 1 hour for astrocytic labeling. Following that, samples were washed for 5 min with PBS and incubated with the secondary antibody Alexa Fluor 488 (goat anti-rabbit IgG, Life Technologies, 1:200) for 2 hours. In the end, samples were washed again with PBS for 5 min and the slides were covered with Fluoroshild with DAPI (Sigma, F6057).

Images of the hippocampal regions granular layer of the dentate gyrus (GrDG), CA1 and CA3 were acquired using the Leica DMi8 inverted microscope and analyzed in the LasX software (Leica Microsystems). The filter used for markings made with NeuN was GFP with excitation at 488 nm and emission at 510 nm. The filter used for GFAP with excitation 561 nm and 594 nm emission. To visualize the cell nucleus, filter for DAPI with 345 nm excitation and 460 nm emission.

Images of six sections of each animal's brain were acquired with a 10x objective lens in the region between -1.70 mm to 2.46 mm, starting from bregma [24]. Quantification was performed in that region using imageJ software 1.51j8 (National Institutes of Health, Bethesda, MD, United States).

**Evaluation of the density of senile plaques.**   For the analysis of the presence and density of senile plaques, thioflavin S was used. Slides with 20 μm brain slices were warmed for 10 min to room temperature and washed with PBS containing 0.1% Triton X-100 (PBST) for 3 min. This process was repeated three times. After that, slides were immersed in 0.1% thioflavin S (Sigma T1892) diluted in PBST for 5 min. Following that, three washes for 3 min with PBS were done followed by immersion in 70% alcohol for 5 min. After a new wash, slides were closed with Fluoroshild with DAPI (Sigma, F6057) and stored at 4˚C. Positive labelling signal was differentiated from the background by visualization in higher magnification and by the experience of the operator.

Images of hippocampal areas CA1, CA3 and the granular layer of the dentate gyrus (GrDG) were obtained in an inverted microscope (Leica, DMi8) and analyzed using LasX software (Leica Microsystems).

## Statistical analysis

Data were expressed as means ± SEM (standard error of means) and analyzed with the Graph Pad Prism program (GraphPad Software, San Diego, CA, version 6). For the Barnes maze test, data from SAMR-1 and SAMP-8 groups were compared. After that, SAMP-8 and lithium-treated SAMP-8 groups were compared. Results from this test were analyzed using a two-way analysis of variance (repeated measures) followed by Bonferroni's multiple comparison test. In molecular and tissue analysis, the homogeneity of sample variance was tested using D'Agostine and Pearson omnibus variety test and Shapiro Wilk normality test. As long as one of the tests indicated no variance among groups, one-way ANOVA was performed followed by Tukey's multiple comparison test.

In all analyses only probability values (*P*) less than 0.05 were considered statistically significant.

## Results

### Behavioral experiments

**Chronic microdose lithium treatment preserved memory of SAMP-8.**   Before the beginning of treatments, at 2 months of age, SAMR-1 (n = 11) and SAMP-8 (n = 24) were submitted to the Barnes maze to verify their spatial memory. During the five days of the "learning period", a significant difference in the behavior between both groups was recorded [$F_{(1,155)}$ = 91.28, p < 0.0001] which was reflected in a significant interaction among data along the time of observation [$F_{(4,155)}$ = 8.70, p < 0.0001]. SAMR-1 learned where the escape box was, as a significant reduction of 82.5% [$F_{(8,32)}$ = 3.43, p = 0.0058] was observed in the time to find this escape box on the 5[th] day when compared to the 1[st] day in the maze (Fig 2A). However, SAMP-8 did not succeed in entering the escape box during the established time, having a significant increase of 1.4 fold in time to enter the escape box on the 5[th] day, when compared to the 1[st] day [$F_{(23,92)}$ = 1.80, p = 0.0256] (Fig 2A). The number of wrong holes explored before entering the right one was also significantly different between both groups along the "learning period" [$F_{(1,155)}$ = 47.34, p < 0.0001]. SAMP-8 group explored significant more holes before finding the right one along the 5 days of the "learning period" [$F_{(23,92)}$ = 10.33, p < 0.0001], whereas SAMR-1 had no difficulties to learn where the escape box was [$F_{(8,32)}$ = 1.01, p = 0.4507] (Fig 2D).

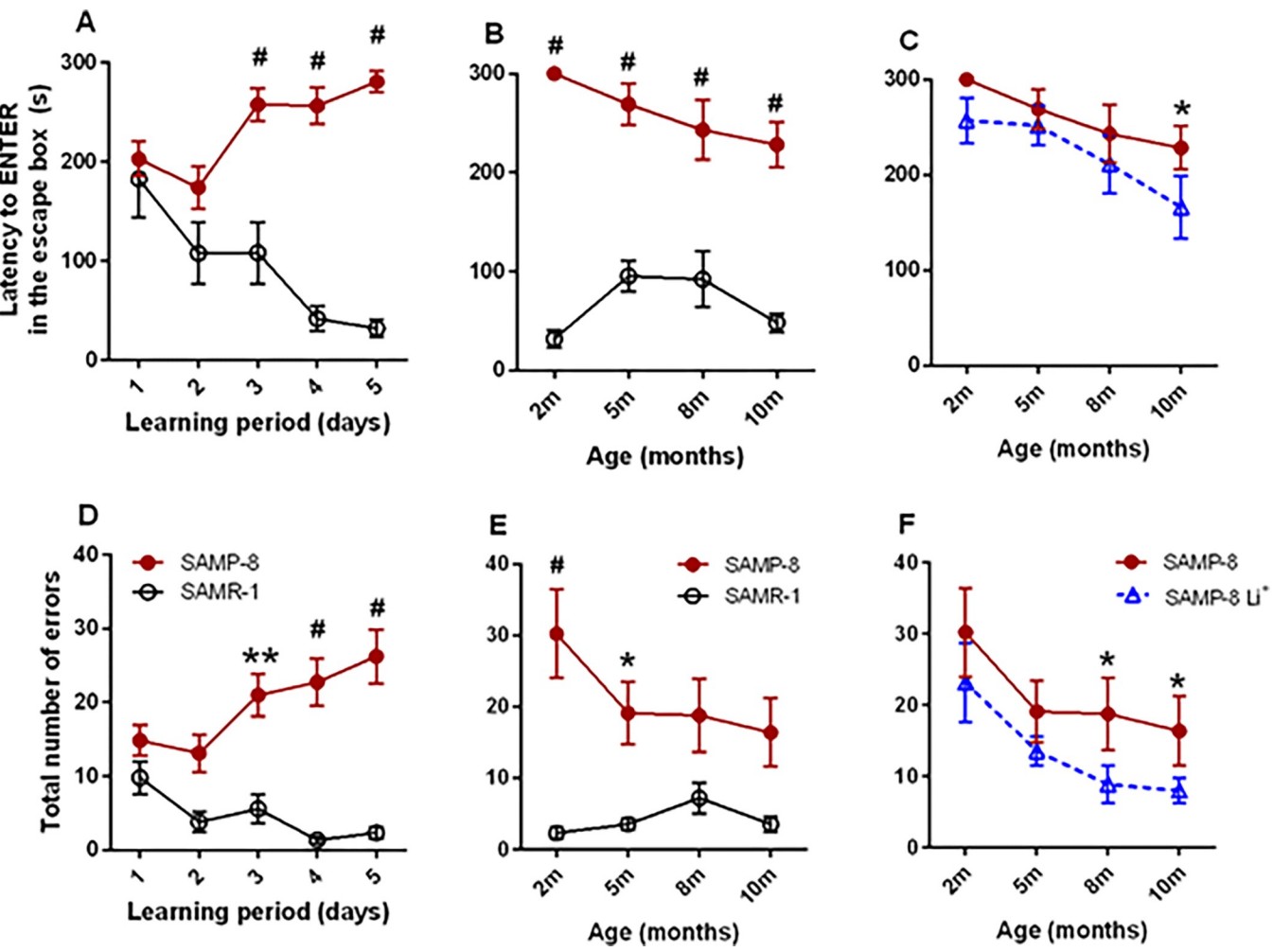

**Fig 2. Effects of treatment with low dose lithium to spatial memory of SAMR-1 and SAMP-8.** Spatial memory was evaluated in the Barnes maze and the time to enter the escape box (A-C) and the number of wrong holes explored (D-F) were recorded. Graphs A and D show the observation during learning time of groups SAMR-1 (n = 11) and SAMP-8 (n = 24), as indicated by the legend in D. Along the mice aging process, the behaviour of non-treated SAMP-8 was compared to SAMR-1 (B and E, as indicated by the legend in E) and SAMP-8 Li⁺ (C and F, as indicated by the legend in F). Along the 8 months' period of observations, some animals have died (for different reasons) and the final number of animals in each group was 9 animals in the SAMR-1, 9 animals in the SAMP-8 and 7 animals in the SAMP-8-Li group. Symbols and vertical bars are means ± SEM. *: $p < 0.05$; **: $p < 0.01$; #: $p < 0.0001$.

After the "learning period", the SAMP-8 group was randomly divided (n = 12 in each group) and the treatments began. Female SAMP-8 were treated with microdose lithium for 8 months (from 2 months until 10 months of age). The animals were resubmitted to the test at 5, 8 and 10 months of age. The difference in time to enter the escape box between SAMR-1 and SAMP-8 was maintained along with all expositions of mice to the Barnes maze [$F_{(1,17)}$ = 174.60, p < 0001] (Fig 2B). The same was observed concerning the number of holes explored before the animal gets to the escape box, as SAMP-8 explored a significantly greater number of holes until they complete 5 months of age [$F_{(1,68)}$ = 36.58, p < 0.0001]. This difference between SAMR-1 and SAMP-8 became not significant in the next two evaluations when animals were 8 and 10 months old (Fig 2E).

The treatment of SAMP-8 with lithium promoted a significant difference in behaviour [$F_{(1,74)}$ = 5.08, p = 0.0272] that was also significant along time [$F_{(3,74)}$ = 4.29, p = 0.0076] (Fig 2C), as SAMP-8-Li showed significant less time to enter in the escape box (Fig 2C) and also

explored significantly less holes to find the escape box [$F_{(1,70)}$ = 5.68, p = 0.0199] along the repeated expositions [$F_{(3,70)}$ = 4.17, p = 0.0089] (Fig 2F), suggesting that they kept the memory of the escape box place. In comparison with SAMR-1 group, treatment with lithium made the SAMP-8 treated group (SAMP-8 Li$^+$) make as fewer errors as SAMR-1 since 8 months of age, showing the efficacy of this long-term treatment.

**Chronic microdose lithium treatment reduced the anxiety of SAMP-8.** Concerning the evaluation of anxiety behavior, all animals were submitted to tests when they were 10 months old. In the elevated plus-maze, SAMP-8 presented a significant decrease of 33.8% in percentage of entries in the open arms (t = 2.83, dF = 16, p = 0.0120, Fig 3A) and also a significant decrease of 77.7% in time spent in open arms (t = 9.02, dF = 16, p < 0.0001, Fig 3B) when compared to SAMR-1, suggesting an increase in anxiety-like behavior. Chronic lithium treatment made no difference in the percentage of entries but increased the time spent in open arms (t = 2.32, dF = 14, p = 0.0357, Fig 3B). Concerning the head-dippings, a significant decrease of 83.5% in the number of movements made by SAMP-8 was observed, when compared to SAMR-1 (t = 4.87, dF = 16, p = 0.0002, Fig 3C), whereas the treatment with lithium significantly increased the behaviour of SAMP-8 (t = 3.41, dF = 14, p = 0.0043, Fig 3C), suggesting a decrease in anxiety-like behavior.

**Low-dose lithium preserved molecular markers related to memory and neuroplasticity.** To verify if chronic low-dose lithium changed the density of proteins in the hippocampus related to long-term memory formation, western-blot analysis for LTP markers was done in samples from the three groups. Initially, it was verified that the SAMP-8 group presented a significant reduction of 73.0% in the density of NMDA receptors compared to SAMR-1 samples (p = 0.0438). Chronic treatment with lithium maintained the density of those receptors as an increased density of 3.8 fold was detected when compared to non-treated SAMP-8, which made SAMP-8-Li comparable to SAMR-1 (Fig 4A). Regarding the other glutamate receptor AMPA, although a slightly similar profile was observed, no significant difference was observed among groups (Fig 4B). The decrease in density of NMDA receptor was followed by a decrease in its anchoring protein, PSD95, as in SAMP-8 mice samples it was verified a significant reduction of 46.7% in the density of this protein when compared to SAMR-1 ones (p = 0.0265). Treatment with lithium maintained the density of PSD95 similar to that observed in SAMR-1 samples and significantly higher in 2.3 fold when compared to SAMP-8 samples (p = 0.0019 (Fig 4C). As a consequence of activation of NMDA receptors and increase in intracellular calcium concentration, there is an increase in activation of CAMK-IV. With the decrease in the density of NMDA receptors, SAMP-8 mice also showed a significant decrease of 37.7% in the

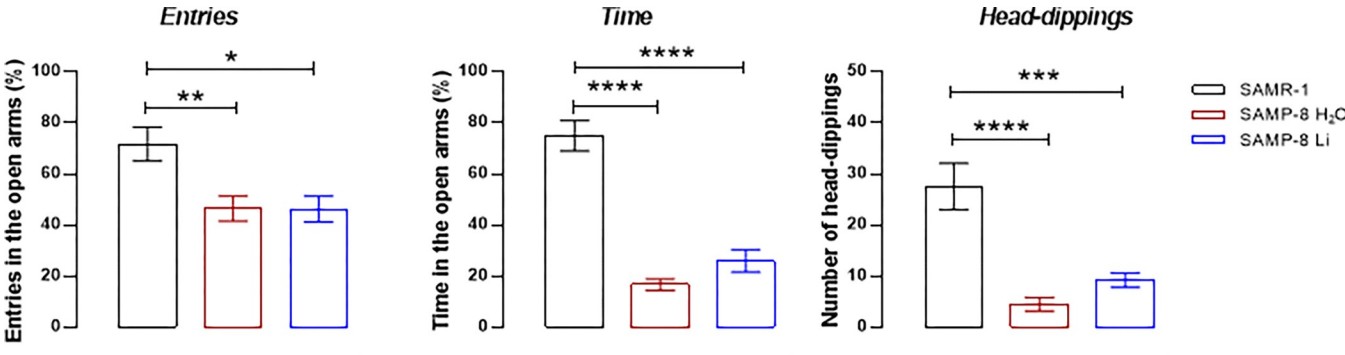

**Fig 3. Effects of treatment with low dose lithium on anxiety-like behavior of mice.** The behavior was evaluated in the elevated plus maze as % of entries (A) and percentage of time (B) number in the open arms and head-dippings (C). Symbols are individual points and histograms and vertical bars are means ± SEM. *: p < 0.05; **: p < 0.01; ***: p < 0.001; ****: p < 0.0001. SAMR-1 (n = 9), SAMP-8 (n = 9), SAMP-8-Li (n = 7).

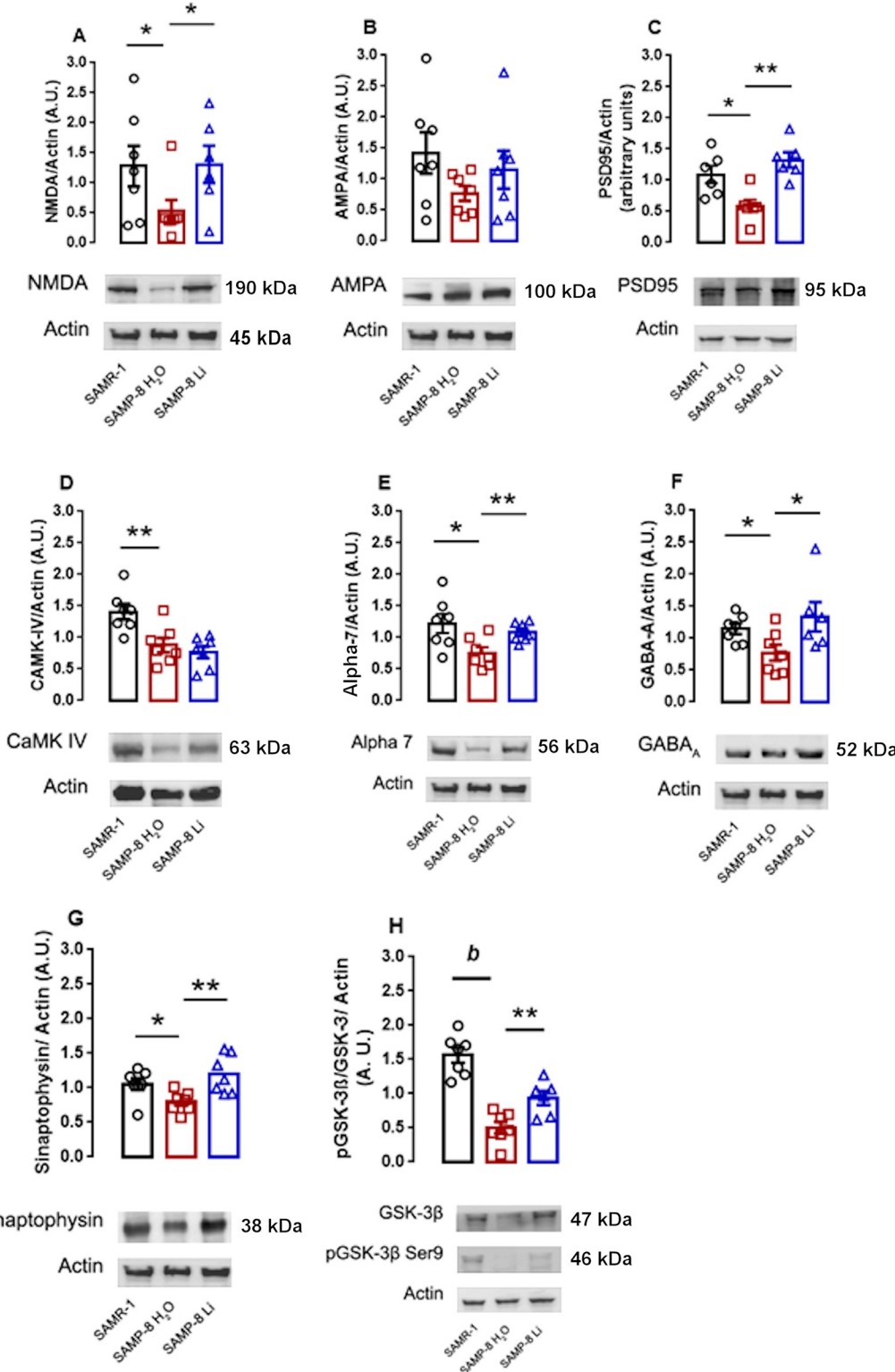

**Fig 4. Effects of chronic treatment with low dose lithium for the density of proteins involved in long-term memory maintenance.** Histograms and vertical bars are means ± SEM. *: p <0.05; **: p <0.01; *b*: p < 0.0001. The SAMR-1 group is represented in black, the SAMP-8 group is represented in red and SAMP-8-Li group is represented in blue. For these experiments 6 to 7 animals of each group were randomly assigned.

density of CAMK-IV (p = 0.0087) compared to SAMR-1. However, chronic treatment with lithium did not change the density of CAMK-IV in hippocampus samples, and SAMP-8-Li animals also showed a significant reduction of 45.5% in CAMK-IV density compared to SAMR-1 samples (p = 0.0019). This means that low-dose lithium treatment did not influence the expression and density of this protein (Fig 4D).

To verify if other neurotransmitter systems that modulate memory maintenance also changed in the SAMP model, α7 nicotinic acetylcholine receptors (nAChR) and GABA$_A$ receptors' densities were analyzed. In SAMP-8 animals, it was verified a significant decrease of 38.1% in α7 nAChR when compared to SAMR-1 samples (p = 0.0166). Animals treated with lithium presented a non-significant increase of 1.43 fold in the density of this receptor (p = 0.1069) and this was not different from SAMR-1 animals either. (Fig 4E). With GABA$_A$ receptors, SAMP-8 presented a non-significant decrease of 32.7% in the density of this receptor when compared to SAMR-1 samples (p = 0.1946). SAMP-8 treated with lithium presented a significant increase of 1.73 fold in the levels of GABA$_A$ receptors compared to untreated SAMP-8 (p = 0.0448) and became comparable to SAMR-1 levels (Fig 4F).

To analyse the density of synaptic terminals in the three groups, synaptophysin was quantified. It was verified a non-significant decrease of 24.4% in the density of this protein in SAMP-8 samples, when compared to SAMR-1 group (p = 0.1009). Treatment with lithium significantly increased in 1.5-fold the density of this protein in SAMP-8-Li samples compared to untreated SAMP-8 (p = 0.0090). This density was even higher than the one detected in SAMR-1 samples (1.13 fold), but this was not statistically significant (Fig 4G).

When it comes to the classical mechanism of action of lithium in the cells, i.e., phosphorylation of GSK-3β in Ser9, it was verified a significant decrease of 67.8% in the phosphorylation of this protein in SAMP-8, when compared to SAMR-1 (p < 0.0001). Treatment with microdose lithium promoted a significant increase of 1.87 fold in GSK-3β phosphorylation when compared to untreated SAMP-8 (p = 0.0201). However, SAMP-8 Li still presented a significant decrease of 39.9% in GSK-3β phosphorylation when compared to SAMR-1 samples (p 0.0013) (Fig 4H).

**Effects of chronic microdose lithium treatment on neuronal, astrocytic and senile plaques density.** The density of neurons and astrocytes was analyzed in hippocampal areas CA1, CA3 and the granular layer of the dentate gyrus (GrDG). In CA1 there was no difference in neuronal density when comparing tissue samples from SAMR-1 and SAMP-8 (t = 1.69, df = 8, p = 0.1289). Chronic treatment with lithium in microdoses slightly increased the neuronal density in SAMP-8 samples (t = 2.28, df = 8, p = 0.0516, Fig 5). Nevertheless in CA3 (Fig 6) and in GrDG (Fig 7) areas the density of neurons in SAMP-8 samples was significantly lower (CA3: t = 3.02, df = 8, p = 0.0165; GrDG: t = 3.30, df = 8, p = 0.0108) when compared to the density of neurons in the same areas in SAMR-1 samples. Treatment with lithium along the aging process maintained the density of neurons in CA3 and GrDG similar to those from the SAMR-1 group, and significantly higher when compared to untreated SAMP-8 samples (CA3: t = 4.64, df = 8, p = 0.0017; GrDG: t = 5.50, df = 8, p = 0.0006, Figs 6 and 7).

Concerning the density of astrocytes, no difference between SAMR-1 and SAMP-8 was observed in the three hippocampal areas analyzed. In the same way, long-term treatment with microdose lithium did not change the astrocytes' density as well (Figs 5–7).

About the senile plaque density, one of the hallmarks of Alzheimer's disease, a significant increase of 4.19 fold was observed in SAMP-8 samples, when compared to SAMR-1 (t = 3.45, df = 8, p = 0.0087). Animals treated with lithium presented a significant reduction of 54.9% of this density (t = 2.45, df = 8, p = 0.0398, Fig 8).

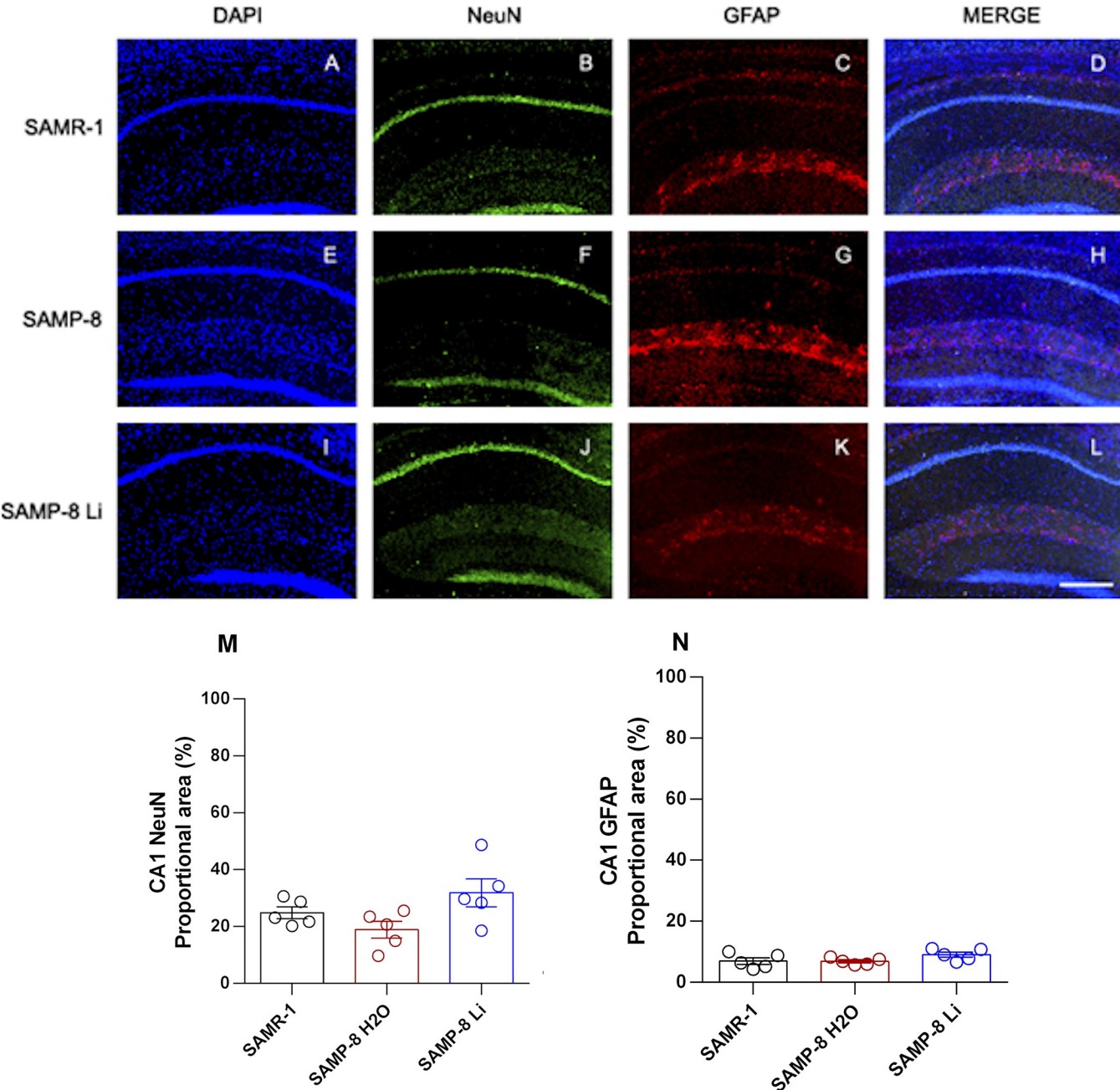

**Fig 5.** Labeling of NeuN (green) and GFAP (red) in the CA1 hippocampal region of SAMR-1 (A–D), SAMP-8 (E–H) and SAMP-8 Li (I–L). Cellular nuclei were labeled with DAPI (blue). Images were obtained using an inverted DMi8 microscope (Leica, Wetzlar, Germany) with a 10× objective. Scale bar: 100μm. M and N: Percentage of NeuN and GFAP labeling in the CA1 area of the three groups. Histograms and vertical bars are means ± SEM of samples from 5 animals per group. *: $p < 0.05$.

## Discussion

Alzheimer's disease (AD) is still the most important cause of dementia worldwide, but until today that is no cure for this devastating disease. In the last years, our research team showed the beneficial effects of the treatment with microdose lithium for memory maintenance and neuroprotection in a transgenic mice model of AD [14], but the cellular mechanisms of this low dosage of lithium are still lacking.

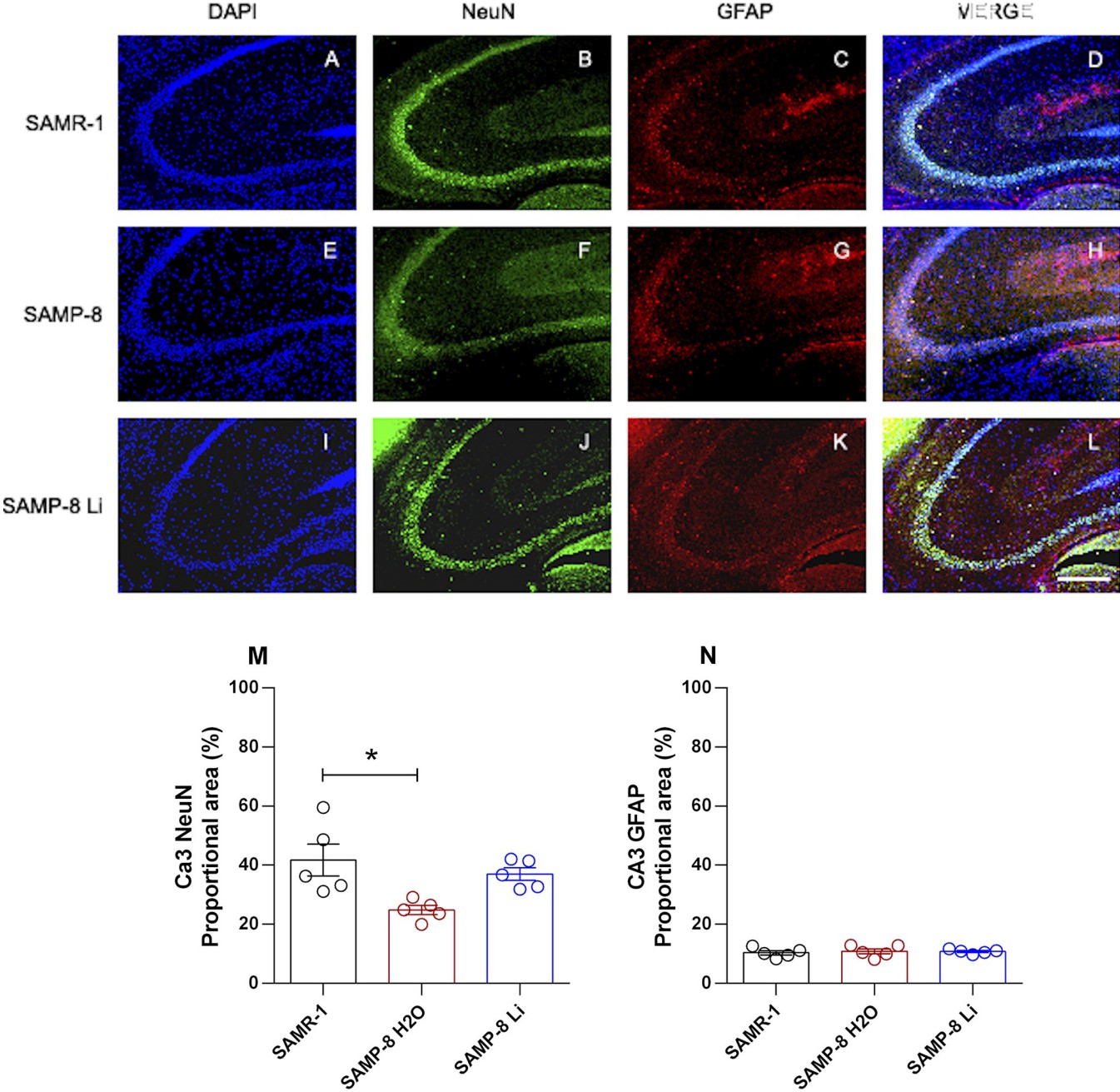

**Fig 6.** Labeling of NeuN (green) and GFAP (red) in the CA3 hippocampal region of SAMR-1 (A–D), SAMP-8 (E–H) and SAMP-8 Li (I–L). Cellular nuclei were labeled with DAPI (blue). Images were obtained using an inverted DMi8 microscope (Leica, Wetzlar, Germany) with a 10× objective. Scale bar: 100μm. M and N: Percentage of NeuN and GFAP labeling in the CA3 area of the three groups. Histograms and vertical bars are means ± SEM of samples from 5 animals per group. *: p < 0.05.

Microdose lithium is being considered as a disease-modifying strategy both in human and animal observations (Nunes et al., 2013, 2015a; Relaño-Ginés et al., 2018), which turns it into a hope for treatment of these diseases, along with lifestyle changes, like the practice of physical activity and the environmental enrichment, strategies that protect the brain from the development of AD and others neurodegenerative diseases (Toricelli et al., 2020; Malerba et al., 2021). In previous works our research team showed that effects of this low dosage are seen when a

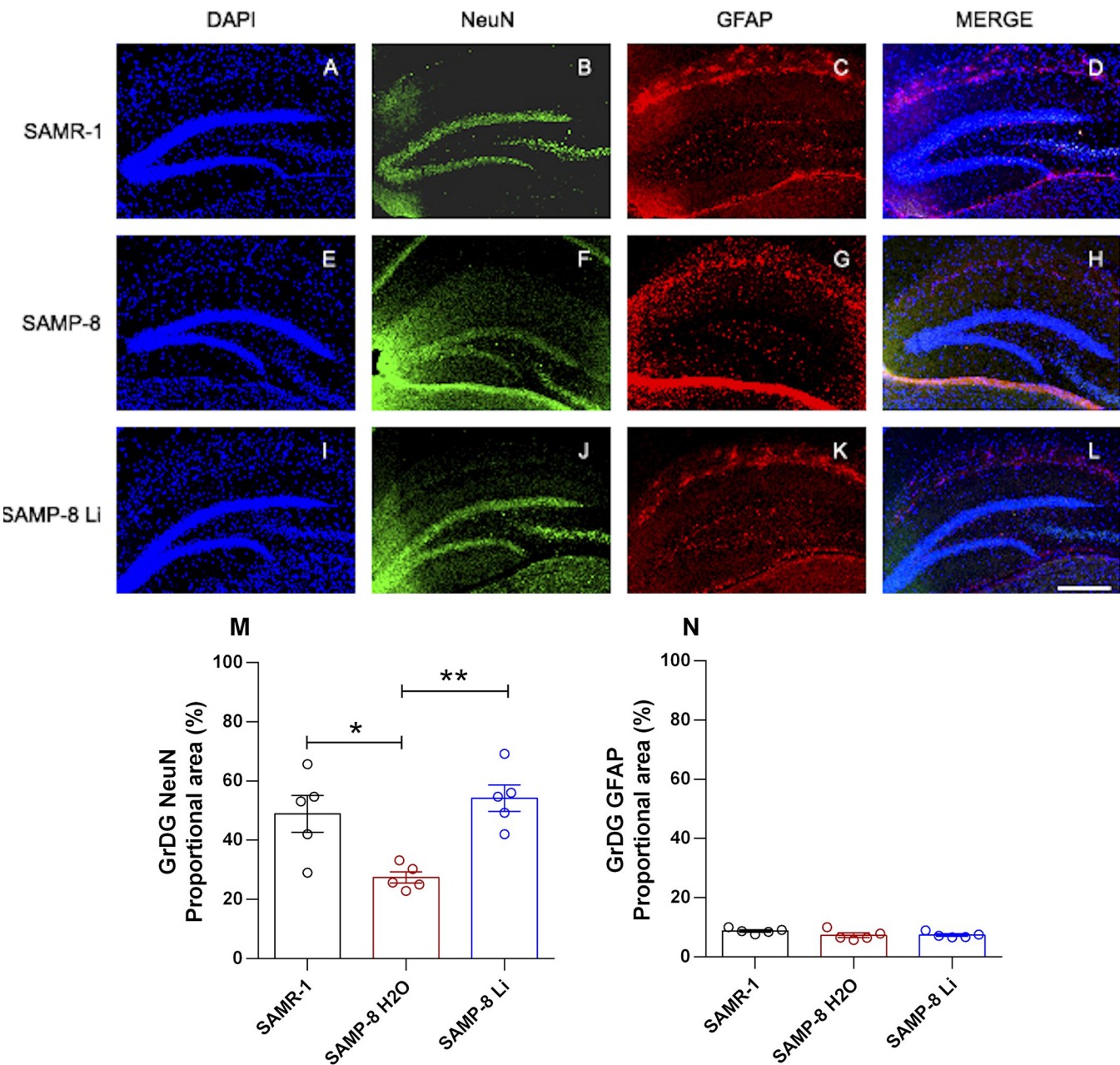

**Fig 7.** Labeling of NeuN (green) and GFAP (red) in the GrDG hippocampal region of SAMR-1 (A–D), SAMP-8 (E–H) and SAMP-8 Li (I–L). Cellular nuclei were labeled with DAPI (blue). Images were obtained using an inverted DMi8 microscope (Leica, Wetzlar, Germany) with a 10× objective. Scale bar: 100μm. M and N: Percentage of NeuN and GFAP labeling in the GrDG area of the three groups. Histograms and vertical bars are means ± SEM of samples from 5 animals per group. *: p < 0.05; **: p < 0.01.

physiological/pathological disturbance is present, as is observed in AD mice models [14]. In this way, there was no effect of low dosage lithium in wild-type mice, unless for an anxiety reduction (Nunes et al., 2015). For this reason, in the present work, we investigated the effects of microdose lithium in SAMP-8, a mice model of accelerated aging, that is also being accepted as a model of dementia (Tomobe and Nomura, 2009; Akiguchi et al., 2017). In the present study, we only used female mice. Many studies well documented the higher prevalence of Alzheimer's disease in women compared to men (for examples, please verify [25–30]. The

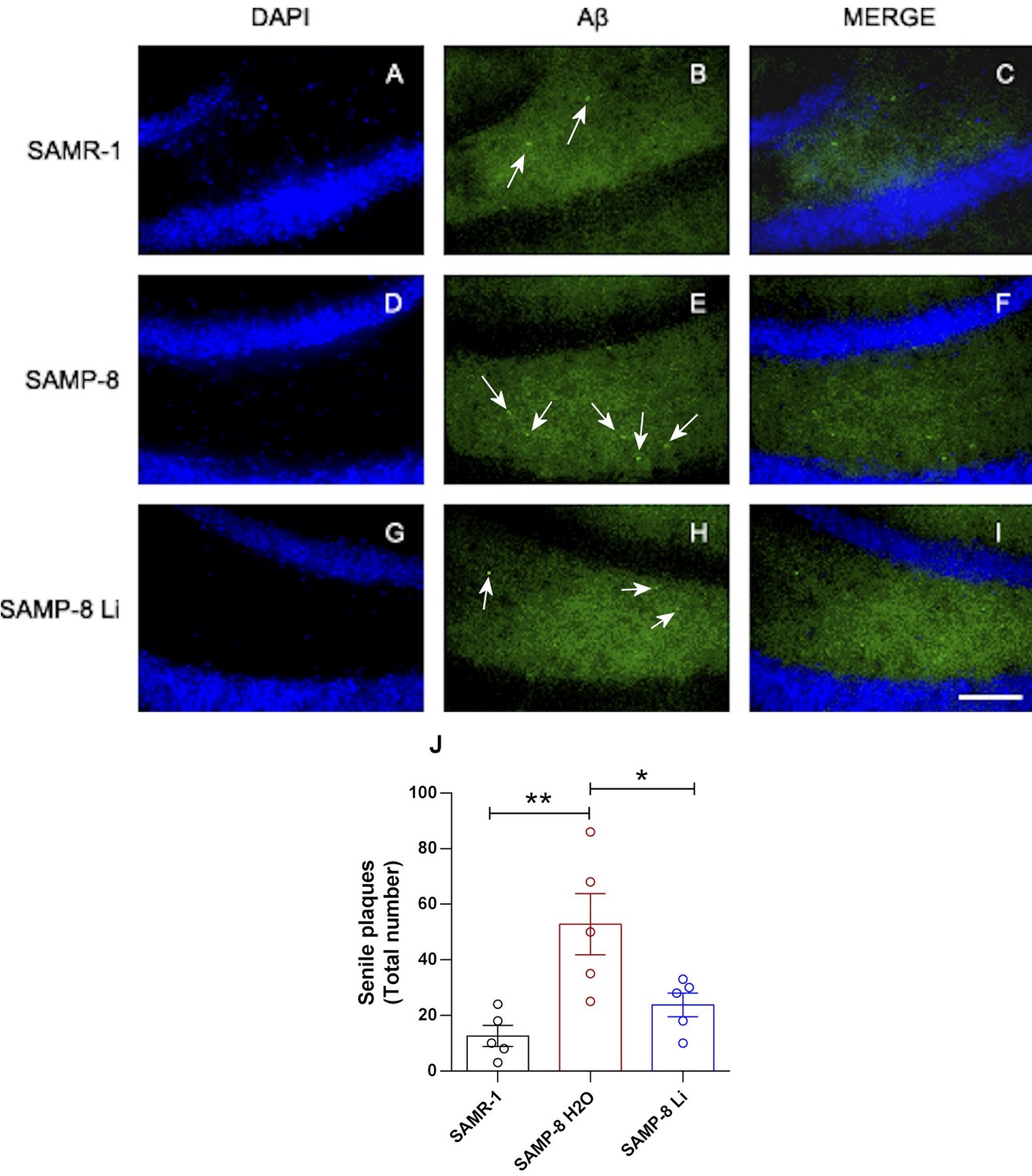

**Fig 8. Senile plaques in the different animals' hippocampus.** Labeling was done with Thioflavin-S 1% (green) in the hippocampus of SAMR-1 (A–C), SAMP-8 (D–F) and SAMP-8 Li (G–I). Cellular nuclei were labeled with DAPI (blue). Images were obtained using an inverted DMi8 microscope (Leica, Wetzlar, Germany) with a 20× objective. Scale bar: 100μm. J: Quantification of the total number of plaques in the hippocampus. Histograms and vertical bars are means ± SEM of samples from 5 animals per group. *: $p < 0.05$; **: $p < 0.01$.

prevailing hypothesis attributes this gender-specific discrepancy to the longer life expectancy of women, when compared to men. Similar to the observation made in male SAMP-8 and transgenic mice for AD in previous works (Nunes et al., 2015a; Malerba et al., 2021), female SAMP-8 also showed a reduction in learning skills as well as in long-term memory, when compared to their natural control SAMR-1, although some molecular mechanisms for the lithium protection were different in female and male animals. In other mice strains the age-related decline in estrogen sensitivity in their tissues, has been demonstrated to relate to exacerbated age-related pathologies as memory decline [31,32].

## Effects on molecular mechanisms involved in memory formation

The behavioural observation with lithium treatment was corroborated with the analysis of proteins involved with the most accepted biological model of long-term memory formation, the long-term potentiation. In the present study, it was clear that those mechanisms are disrupted in female SAMP-8, which explains the low capacity of these animals to learn and memorize tasks. Interestingly, in previous work from our research group, no difference in NMDA receptors density was observed in hippocampus homogenates of male SAMP-8 mice when compared to SAMR-1 at 10 months of age [13]. In the present work, the density of both NMDA and PSD95 in the female SAMP-8 hippocampus were lower when compared to SAMR-1 samples. Although not measured in the present work, the relevance of female hormones as estrogen to NMDA binding in rodents' hippocampus was already described [33–35]. In SAMP-8 animals treated with lithium, the increased density of NMDA receptors and PSD95 corroborates to the action of lithium in the LTP signaling mechanism mediated by glutamate. PSD95 is an important scaffolding protein in the glutamatergic synapse, mediating interactions between signaling proteins, such as AMPA and NMDA receptors (Sheng and Kim, 2011). In addition, it plays a fundamental role in the structure of excitatory synapses, binding the receptors to cytoskeletal proteins and mediating synaptic morphology and receptor traffic (De Wilde et al., 2016). Although no difference in the density of AMPA receptors was seen, the increase in NMDA receptors and PSD95 in lithium-treated animals may indicate an increase in efficiency and synaptic strength in glutamate-mediated signaling.

## Effects on neuronal and synapses survivor and senile plaques' densities

The preservation of neuronal density is a typical characteristic of the treatment with the lithium in low-dose, as we have been observed this effect in the hippocampus of 16-months-old transgenic mice model for AD [14] and 10-months-old male SAMP-8 [13] with the chronic treatment with lithium. Also the treatment of *ex-vivo* hippocampus slices with lithium in low-dose also evidenced the neuroprotection observed *in vivo* [36]. In the present work, the same neuroprotective effect was observed when female SAMP-8 were treated with the same lithium dose. This show a consistent effect of this treatment along the aging process of mice that reflects the maintenance of memory as observed in humans treated with similar doses [15]. Interestingly no effects on astrocytes (as observed with GFAP labeling) was observed with this treatment.

Increases in senile plaques are a hallmark of AD. Although some groups have not detected senile plaques in the mouse model used in the present work, our research team and others showed previously that Thioflavin S is able to stain senile plaques in SAMP-8, as evidenced by the observed fluorescence in brain tissue from these animals [13,37]. These newer investigation, including our own, suggest that Thiflavin S is capable of staining senile plaques in SAMP-8. Besides disrupting neuronal transmission, Aβ plaques can inactivate NMDA receptors, compromising the intracellular cascade for memory consolidation (Yin et al., 2015). In our

study, it was observed a decrease in the density of CAMK-IV in SAMP-8, a $Ca^{2+}$-dependent protein kinase that is part of the long-term memory formation pathway (Ahmed et al., 2000). SAMP-8 animals treated with lithium presented a significant decrease in senile plaques with the increase in expression of NMDA receptors density without, however, a change in CAM-K-IV density, which means that the neuroprotective mechanism of lithium in memory formation does not necessarily affects all the proteins in that molecular pathway.

Another synaptic protein that was maintained with the lithium treatment was the synaptophysin. This was already observed in a transgenic model of AD treated with the same lithium dose, together with the increase in the density of the brain-derived neurotrophic factor (BDNF) (Nunes et al., 2015a). Corroborating with these observations and others made with the same microdose lithium in other AD animal models (Toricelli et al., 2020; Malerba et al., 2021), lithium treatment maintained the density of neuronal cells in female SAMP-8 animals, which reinforces its role in neuroprotection as mentioned by other groups in experimental and clinical observations (Palmos et al., 2021; Puglisi-Allegra et al., 2021).

### Effects on α7 nicotinic acetylcholine and GABA$_A$ receptors

Many neurotransmitter systems contribute to memory maintenance, including the cholinergic system which has a prominent role in the induction and modulation of LTP. Most studies show the loss of nicotinic receptors in AD patients, especially the α7 subtype (α7nAChR) that contributes to the modulation of long-term memory formation (Hampel et al., 2018; Alkadhi, 2021). Upholding of this system is one of the strategies for delaying the advance of AD in the initial and moderate phases of the illness with the use of cholinesterase inhibitors (Athar et al., 2021). In presynaptic neurons, the α7 nicotinic receptor induces the synthesis and release of glutamate, an essential neurotransmitter involved in neuroplasticity (Lozada et al., 2012; Haam and Yakel, 2017). In postsynaptic neurons, α7 nicotinic receptor activation induces the activity of CaMK-IV and transcription factors such as CREB. Recently our group showed that inhibition of the α7 nicotinic receptor prevented memory retrieval in mice submitted to an experimental model of neurodegeneration (Telles-Longui et al., 2019). In the present work, SAMP-8 presented a significant decrease in the density of α7nAChR when compared to SAMR-1, which may be related to the critical observed memory loss. However, lithium-treated SAMP-8 mice presented no loss in the density of this receptor, which could be a significant fact in memory maintenance.

Another receptor that was preserved with lithium treatment in the SAMP-8 hippocampus was the GABA$_A$ receptor. This receptor is implicated both in memory maintenance and in anxiety mechanisms. Some works already pointed out the relationship between memory loss and the development of anxiety and depression. These conditions, if not observed and treated with care, may turn into risk factors for developing AD in old age (Sinoff and Werner, 2003; Kassem et al., 2017).

Lithium carbonate has been considered a successful strategy to control humor changes in bipolar human disorder. Other recent pieces of evidence pointed to the potential role of the metal lithium as an anxiolytic agent combined or not with other anxiolytic drugs (Walia et al., 2019) with implications for the mania phase of bipolar disorder (Chaves Filho et al., 2020), as well as to neuroprotection in the neurodegenerative process (Nunes et al., 2015b; Won and Kim, 2017; Morlet et al., 2018; Habib et al., 2019; Malerba et al., 2021). In the present work, long-term treatment with microdose lithium reduced the anxiety-like behavior and increased the density of GABA$_A$ receptors, which may also have contributed to memory maintenance along the aging process of SAMP-8.

### Inhibition of the GSK-3β

One of the neuroprotective mechanisms attributed to lithium involves the direct inhibition of the enzyme glycogen synthase kinase 3β (GSK-3β) (Klein and Melton, 1996). The inhibition of GSK-3β through phosphorylation in ser9 promotes the maintenance of synaptic function and axonal transport and decreases apoptosis and inflammatory markers mediated or not by NF-κB (Llorens-Martín et al., 2014; Toricelli et al., 2020).

In the present study, it was shown that SAMP-8 animals treated with lithium had an increase in GSK-3β inhibition when compared with SAMP-8 animals without treatment, confirming the activation of the classical mechanism. Besides, lithium-treated animals presented a reduction in the density of senile plaques. A previous study using human and mouse neuroblastoma cell culture suggests that inhibition of GSK-3β by AR-A014418 decreases APP cleavage by BACE1, reducing the production of Aβ peptides. This regulatory action of GSK-3β in the production of Aβ seems to be mediated by NF-κB, where the decrease in the signaling of this pathway reduces the gene expression of BACE1 (Ly et al., 2013). Corroborating with this information, in a previous study, our group showed that microdose lithium decreased the density of NF-κB and other inflammatory markers in the organotypic hippocampal culture of SAMP-8 animals (Toricelli et al., 2020).

In conclusion, chronic microdose lithium seems to act as a long-term memory and neuronal protector, promoting functional and structural maintenance of several pathways compromised in AD. The maintenance of memory observed in this and previous studies by our research team put pieces of a puzzle together to show that there is no unique mechanism capable of explaining the beneficial effects of this treatment but rather a combination of multiple neuroprotective mechanisms. Increases in BDNF levels, observed in male SAMP-8 [13] and in transgenic mice model for Alzheimer's disease [14], synaptic and neuronal maintenance in hippocampal areas of CA1, CA3 and GrDG, as well as inhibition of GSK-3β (all seen in the present work), reduction of neuroinflammation observed in SAMP-8 *ex vivo* hippocampus [36] and reduction of senile plaques (also observed in this and previous works), are some of the observed effects that may, altogether, contribute to long-term behavioral observations. Also, changes seen in this study in GABA$_A$ and α7 nicotinic receptors suggest an essential role of lithium in maintaining and restoring the balance in the formation of memory and reduction

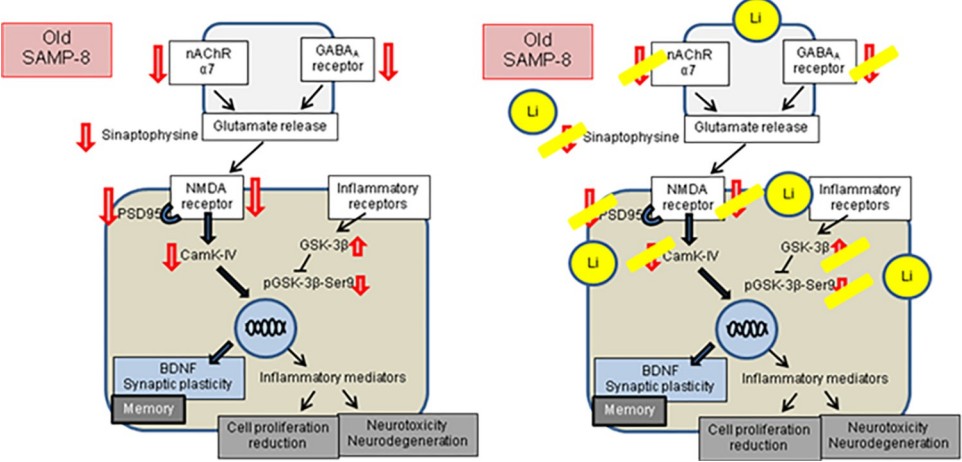

**Fig 9. Global effects of treatment with microdose lithium for LTP.** Many steps in the LTP formation were disrupted in SAMP-8 mice, while chronic treatment with lithium in microdose avoided the decrease in many protein densities, allowing memory maintenance in older mice.

of anxious behavior. Therefore, this work brings a necessary contribution to the understanding of the mechanisms of microdose lithium treatment along the aging process to promote neuronal protection, memory maintenance, and reduction of anxiety, as abstracted in Fig 9.

## Supporting information

**S1 File. Images of original membranes with antibody and Ponceau reagent markings.** (PDF)

## Author Contributions

**Conceptualization:** Tania Araujo Viel.

**Data curation:** Mariana Toricelli, Hudson Sousa Buck.

**Formal analysis:** Arthur Antonio Ruiz Pereira, Alessandra Macedo Pinto, Helena Nascimento Malerba, Tania Araujo Viel.

**Funding acquisition:** Tania Araujo Viel.

**Investigation:** Arthur Antonio Ruiz Pereira, Helena Nascimento Malerba, Mariana Toricelli.

**Methodology:** Arthur Antonio Ruiz Pereira, Mariana Toricelli.

**Project administration:** Tania Araujo Viel.

**Supervision:** Tania Araujo Viel.

**Writing – original draft:** Arthur Antonio Ruiz Pereira, Tania Araujo Viel.

**Writing – review & editing:** Mariana Toricelli, Hudson Sousa Buck, Tania Araujo Viel.

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
