## [Decision Letter · Decision Letter 0]

28 Nov 2023

PONE-D-23-36699Multiple mechanisms of microdose lithium protect behavioral deficits and molecular mechanisms for memory formation in SAMP-8, a mouse model of accelerated aging.PLOS ONE

Dear Dr. Viel,

Thank you for submitting your manuscript to PLOS ONE. After careful consideration, we feel that it has merit but does not fully meet PLOS ONE’s publication criteria as it currently stands. Therefore, we invite you to submit a revised version of the manuscript that addresses the points raised during the review process.

We look forward to receiving your revised manuscript.

Kind regards,

Kyung-Wan Baek, Ph.D.

Academic Editor

PLOS ONE

Journal Requirements:

3. Please remove your figures from within your manuscript file, leaving only the individual TIFF/EPS image files, uploaded separately. These will be automatically included in the reviewers’ PDF.

Additional Editor Comments:

After reviewing the manuscript, the reviewers have recommended a major revision. We urge the authors to carefully consider the feedback provided by the reviewers and make the necessary changes.

Reviewers' comments:

Reviewer's Responses to Questions

**Comments to the Author**

1. Is the manuscript technically sound, and do the data support the conclusions?

Reviewer #1: Partly

Reviewer #2: Yes

Reviewer #3: No

2. Has the statistical analysis been performed appropriately and rigorously? 

Reviewer #1: I Don't Know

Reviewer #2: Yes

Reviewer #3: No

3. Have the authors made all data underlying the findings in their manuscript fully available?

Reviewer #1: Yes

Reviewer #2: Yes

Reviewer #3: No

4. Is the manuscript presented in an intelligible fashion and written in standard English?

Reviewer #1: No

Reviewer #2: Yes

Reviewer #3: Yes

5. Review Comments to the Author

Reviewer #1: Arthur Antonio Ruiz Pereira et al., in their article "Multiple mechanisms of microdose lithium protect behavioral deficits and molecular mechanisms for memory formation in SAMP-8, a mouse model of accelerated aging.," apply chronic treatment with lithium to show its effects on spatial memory, anxiety, and molecular mechanisms related to long- term memory formation during the aging process of a mouse model of accelerated aging (SAMP-8).

Critique:

I. Introduction.

1. The authors used long-term microdose lithium to treat SAMP-8 mice as a model of late-onset AD. The authors admit that the hypothesis is not novel, and lithium was suggested as an anti-AD drug by several groups. The authors should reference the papers related to the topic.

a. Christopher Baethge (2020) stated that low-dose lithium is efficient against dementia (1)

b. Robert Haussmann et al. (2021) suggested that lithium is a therapeutic option in Alzheimer’s disease (2).

c. Shanquan Chen et al. (2022) performed a retrospective cohort study to show the Association between lithium use and the incidence of dementia (3).

2. The novelty of the study is not presented clearly. The authors mention their clinical study with low-dose lithium. Of note, short-term treatment with lithium did not demonstrate promising results (4). However, long-term treatment protocol has been applied already. Orestes V. Forlenza et al. (2019) investigated the clinical and biological effects of long-term lithium treatment in older adults with amnestic mild cognitive impairment. They ran a long-term (24-month-long) randomised clinical trial to show the effects of lithium (5).

Moreover, a systematic review by Sivan Mauer et al. (2014) demonstrated that Lithium, in both standard and trace doses, appears to have biological benefits for dementia (6). Shinji Matsunaga et al. (2015) performed a systematic review and Meta-Analysis of several clinical trials to conclude that lithium treatment may have beneficial effects on cognitive performance in subjects with MCI and AD dementia.

Therefore, we know that micro-doses of lithium (15-300 µg) are efficient against dementia and have little side-effects in long-term protocols.

Consequently, the authors must clearly state the hypothesis and experimental goals while considering animal welfare for long-term experiments.

II. Methods.

1. Treatments. The authors must justify the dose applied and present the calculations. Unfortunately, the reference they provide does not have it.

2. Experimental design. The authors use only females in their study. Please clarify and justify it.

3. Behavioral tests. Barnes maze. The authors apply a very unusual protocol for the Barnes maze, which generally assesses spatial memory acquisition and recall. And here. The authors repeated the task when animals were 5, 8, and 10 months old to assess memory recovery. So, please define the term “memory recovery” and present a reference showing the method.

4. Evaluation of the density of senile plaques with Thioflavin S. The authors used a solution of thioflavin S (Sigma T1892) to stain amyloid plaques in SAMP-8 mice. The same method has been used by numerous labs in the transgenic AD animals. However, aged SAMP-8 mice, in contrast to AD mice, do not present amyloid-beta plaques detectable with thioflavin S (7). Please explain or apply another staining (IHC) for amyloid to show changes.

III. Results.

1. Fig. 2. The authors applied repeated measures ANOVA to show the effects of treatment with low-dose lithium on spatial memory, however, the presentation of the results is not correct. You cannot demonstrate statistical significance with repeated measures ANOVA by the way you did it (at the points of measurements) instead, you must analyze the entire plot and compare the curves.

2. Fig. 8. You are only displaying artifacts, no plaques. Look at reference #7 fig. 3.

IV. Discussion. There is no molecular mechanism explaining the effects of low-dose lithium.

I propose a theory that could be a good fit.

1. In 1986 Adlercreutz et al. showed that Lithium lowers renal, cardiac and splenic ornithine decarboxylase activity (ODC) in mice (8).

2. In 1992 Gad M. Gilad Gad Gilad et al. demonstrated that chronic lithium treatment suppresses ornithine decarboxylase (ODC) activity in the brain (9).

3. Kan et al. (2015) convincingly demonstrated that pharmacologic disruption of the arginine utilization pathway by an inhibitor (DMFO) of arginase and ornithine decarboxylase protects the mice from AD-like pathology (10).

4. Polis et al. (2018) showed that arginase inhibition reverses cognitive decline and synaptic loss in a murine model of Alzheimer’s Disease (11).

Accordingly, to enhance the quality of their work and deliver a unique and noteworthy research paper, the authors must employ additional techniques to investigate the hypothesis based on the findings of various groups about the influence of lithium salts on arginase and ODC expression levels and activities. It might be WB, IHC, or another method. I believe you have the lysates and the brains to run an experiment.

V. It is necessary to improve the quality of scientific English language. This includes correcting any errors in spelling, grammar, and punctuation to make the text clearer and more precise.

References:

1. Baethge, C. Low-dose lithium against dementia. Int J Bipolar Disord 8, 25 (2020). https://doi.org/10.1186/s40345-020-00188-z

2. Haussmann R, Noppes F, Brandt MD, Bauer M, Donix M. Lithium: A therapeutic option in Alzheimer's disease and its prodromal stages? Neurosci Lett. 2021 Aug 24;760:136044. doi: 10.1016/j.neulet.2021.136044. Epub 2021 Jun 10. PMID: 34119602.

3. Chen S, Underwood BR, Jones PB, Lewis JR, Cardinal RN. Association between lithium use and the incidence of dementia and its subtypes: A retrospective cohort study. PLoS Med. 2022 Mar 17;19(3):e1003941. doi: 10.1371/journal.pmed.1003941. PMID: 35298477; PMCID: PMC8929585.

4. Hampel H, Ewers M, Bürger K, Annas P, Mörtberg A, Bogstedt A, Frölich L, Schröder J, Schönknecht P, Riepe MW, Kraft I, Gasser T, Leyhe T, Möller HJ, Kurz A, Basun H. Lithium trial in Alzheimer's disease: a randomized, single-blind, placebo-controlled, multicenter 10-week study. J Clin Psychiatry. 2009 Jun;70(6):922-31. PMID: 19573486.

5. Forlenza, O., Radanovic, M., Talib, L., & Gattaz, W. (2019). Clinical and biological effects of long-term lithium treatment in older adults with amnestic mild cognitive impairment: Randomised clinical trial. The British Journal of Psychiatry, 215(5), 668-674. doi:10.1192/bjp.2019.76

6. Mauer S, Vergne D, Ghaemi SN. Standard and trace-dose lithium: a systematic review of dementia prevention and other behavioral benefits. Aust N Z J Psychiatry. 2014 Sep;48(9):809-18. doi: 10.1177/0004867414536932. Epub 2014 Jun 11. PMID: 24919696.

7. Porquet D, Andrés-Benito P, Griñán-Ferré C, Camins A, Ferrer I, Canudas AM, Del Valle J, Pallàs M. Amyloid and tau pathology of familial Alzheimer's disease APP/PS1 mouse model in a senescence phenotype background (SAMP8). Age (Dordr). 2015 Feb;37(1):9747. doi: 10.1007/s11357-015-9747-3. Epub 2015 Feb 8. PMID: 25663420; PMCID: PMC4320125.

8. Adlercreutz C, Rosengren E, Uvelius B. Lithium lowers renal, cardiac and splenic ornithine decarboxylase activity in mice. Experientia. 1986 Apr 15;42(4):409. doi: 10.1007/BF02118632. PMID: 3007201.

9. Gilad GM, Gilad VH, Wyatt RJ, Casero RA Jr. Chronic lithium treatment prevents the dexamethasone-induced increase of brain polyamine metabolizing enzymes. Life Sci. 1992;50(18):PL149-54. doi: 10.1016/0024-3205(92)90289-2. PMID: 1313939.

10. Kan MJ, Lee JE, Wilson JG, Everhart AL, Brown CM, Hoofnagle AN, Jansen M, Vitek MP, Gunn MD, Colton CA. Arginine deprivation and immune suppression in a mouse model of Alzheimer's disease. J Neurosci. 2015 Apr 15;35(15):5969-82. doi: 10.1523/JNEUROSCI.4668-14.2015. PMID: 25878270; PMCID: PMC4397598.

11. Polis B, Srikanth KD, Elliott E, Gil-Henn H, Samson AO. L-Norvaline Reverses Cognitive Decline and Synaptic Loss in a Murine Model of Alzheimer's Disease. Neurotherapeutics. 2018 Oct;15(4):1036-1054. doi: 10.1007/s13311-018-0669-5. PMID: 30288668; PMCID: PMC6277292.

Reviewer #2: Comments to Authors

In the present manuscript, “Multiple mechanisms of microdose lithium protect behavioral deficits and molecular mechanisms for memory formation in SAMP-8, a mouse model of accelerated aging”, authors examined the effects of microdose lithium on behavioral abnormalities, anxiety, plaque formation and neuroplasticity in SAMP-8 mice. The study is planned and executed well and findings are interesting and relevant to the current research in the area of Alzheimer’s disease. However, the following minor changes are required for the further improvement of the manuscript.

Title

1. The title is vague kindly rewrite it to make it clearer.

2. The word “mechanism” is used twice in the title, please avoid repetition.

Abstract

1. Alzheimer's disease (AD) is the most common affection.

Affection does not seem a suitable word.

Introduction

• Kindly state the objective/s of the study at the end of the introduction section.

Methods

Evaluation of neurons and astrocyte densities in the hippocampus: Page 10

1. Images of the hippocampal regions: GD, CA1, and CA3 were acquired using the Leica DMi8 177 inverted microscope.

What is meant by GD here? Is dentate gyrus (DG)?

Results

1. Figure 2A, B, and C: Provide the figure legend to represent the groups such as SAMR-1 and SAMP-8

2. Figure 2: Compare the results of SAMP-8 Li+ with SAMR-1 to further clarify the effects of lithium microdose.

3. Figure 4D: What could be the possible reason for the decrease of CaMK4 following low-dose lithium treatment? Please explain.

Reviewer #3: This paper addresses the effects of microdose lithium carbonate treatment (2-10 months of age) on Alzheimer-related brain phenotypes in the SAMP-8 mouse model of aging. Cognition was assessed using the Barnes maze longitudinally. Elevated plus maze was used to assess anxiety related behaviors. After treatment, brains were collected and stained for neurons, astrocytes, Thio-s and lysates were used to probe synaptic proteins and GSK-3Bser9 phosphorylation.

While the study is important and well written, there are major issues that call into question the reliability of the data presented.

1. One major issue is that it is impossible to know how much lithium each individual mouse received, as these may vary across the cage. 5 ug/day/animal. Is the entire mixture consumed daily? If ad libitum, this suggests the dose fluctuates.

2. More details on SAMR-1 and SAMP-8 mice are needed. What are these mice? How are they generated?

3. Line 89. There should be a SAMR-1 lithium treated group.

4. Line 152, 153, etc: typo in phosphorylated

5. Line 164: typo

6. Line 183: Concern that this method is not reliable. The detection range should not be altered between slices.

7. Line 184: relations should be relation

8. Line 192: typo

9. Line 204: Overall there are issues with the statistics. 2way ANOVA described in methods (with repeated measures). Looks like in some places only a t-test is used. The methods describe a 2way ANOVA with molecular assays, but there are not enough levels in the SAMR-1, because there was no lithium treated group, so it is unclear what the two-way ANOVA was comparing. P values of 0.05 are listed in the methods but not used if tests used were actually multiple t-tests.

10. Fig 2A, 2D: It is unclear why after day 2, latency to enter escape box and number of errors are worse than baseline for the SAMP-8 mice.

11. Figure 2C: If the SAMP-8 mice were divided to receive Li treatment, why do the SAMP-8 Li+ group already look better? Should have checked that the groups were balanced/equal before continuing with treatment.

12. Fig. 3: All individual data points should be shown.

13. Line 283: typo

14. Line 278: Full uncropped blots should be provided in supplement. Additionally, a single lane from one of each mouse is not sufficient.

15. Figure 4: It is unclear how these differences are tested. It should be one way ANOVA with appropriate post-hoc test. Appear to be multiple t-tests.

16. Line 326: abbreviation “a” appears as “b” in Fig 4H.

17. Line 328: n of 6 in 3C and F but legend says n of 7.

18. Overall, the use of WB for quantitation of synaptic proteins is antiquated because of its low sensitivity. Immunofluorescent staining and confocal imaging of serial sections with appropriate co-stains is more accurate.

19. It appears that the Actin band is re-used in Figs 4A,B,E,F and G. While it might be possible to probe multiple proteins of different molecular weights on the same blot, use secondaries with separate fluorescent channels, or even strip the plot to reprobe, the shape of the bands should be similar. In this case, Fig 4B is concerning because while the Actin bands are catching upward, the AMPA bands are catching downwards, suggesting they are from different gels/blots. The Actin and AMPA bands must be from the same blot. Showing the full blot would clear this up.

20. Fig. 5,6,7: Neuron number cannot be quantified by NeuN percent area. Use stereology.

21. Fig. 8: Should be labeled thio-s not AB. Thio-S can stain many other structures beyond just AB. Moreover, the Thio-s staining is not convincing.

22. Figure 8J: Individual data points from 5 mice should be shown.

23. Line 405: typo.

24. Line 406: this quote should be removed. This idea has since been dismissed. The increased prevalence for AD in women is equal to men at ages where the death rate is equal. Mice, which do not undergo menopause, do however show a lifeline decline in estrogen sensitivity in their tissues and this has been shown to relate to exacerbated pathology.

6. PLOS authors have the option to publish the peer review history of their article (what does this mean?). If published, this will include your full peer review and any attached files.

Reviewer #1: **Yes: **Baruh Polis

Reviewer #2: **Yes: **Sanila Amber

Reviewer #3: No

---

## [Author Response · Author response to Decision Letter 0]

12 Jan 2024

Dear Editor and Reviewers,

We would like to express our sincere gratitude for your valuable feedback and thoughtful comments on our article titled " Microdose lithium improves behavioral deficits and modulates molecular mechanisms of memory formation in female SAMP-8, a mouse model of accelerated aging". We truly appreciate the time and effort you have dedicated to reviewing our work and providing constructive suggestions for improvement.

First and foremost, we would like to address each of the points you raised individually:

Reviewer #1: 

Arthur Antonio Ruiz Pereira et al., in their article "Multiple mechanisms of microdose lithium protect behavioral deficits and molecular mechanisms for memory formation in SAMP-8, a mouse model of accelerated aging.," apply chronic treatment with Lithium to show its effects on spatial memory, anxiety, and molecular mechanisms related to long- term memory formation during the aging process of a mouse model of accelerated aging (SAMP-8).

Critique:

I. Introduction.

1. The authors used long-term microdose lithium to treat SAMP-8 mice as a model of late-onset AD. The authors admit that the hypothesis is not novel, and Lithium was suggested as an anti-AD drug by several groups. The authors should reference the papers related to the topic.

a. Christopher Baethge (2020) stated that low-dose Lithium is efficient against dementia (1)

b. Robert Haussmann et al. (2021) suggested that Lithium is a therapeutic option in Alzheimer's disease (2).

c. Shanquan Chen et al. (2022) performed a retrospective cohort study to show the Association between lithium use and the incidence of dementia (3). 

Answer: We thank the Reviewer for the suggestion. The authors cited the references (lines 53 and 54) together with the explanation of the hypothesis and experimental goals asked in item 2 below.

2. The novelty of the study is not presented clearly. The authors mention their clinical study with low-dose Lithium. Of note, short-term treatment with Lithium did not demonstrate promising results (4). However, long-term treatment protocol has been applied already. Orestes V. Forlenza et al. (2019) investigated the clinical and biological effects of long-term lithium treatment in older adults with amnestic mild cognitive impairment. They ran a long-term (24-month-long) randomised clinical trial to show the effects of Lithium (5).

Moreover, a systematic review by Sivan Mauer et al. (2014) demonstrated that Lithium, in both standard and trace doses, appears to have biological benefits for dementia (6). Shinji Matsunaga et al. (2015) performed a systematic review and Meta-Analysis of several clinical trials to conclude that lithium treatment may have beneficial effects on cognitive performance in subjects with MCI and AD dementia.

Therefore, we know that micro-doses of Lithium (15-300 µg) are efficient against dementia and have little side-effects in long-term protocols.

Consequently, the authors must clearly state the hypothesis and experimental goals while considering animal welfare for long-term experiments.

Answer: We thank the Reviewer for the observation, and we are very happy that there is no doubt about the clinical effects of low-dose lithium. However, this low dose is not in current use by physicians to prevent Alzheimer’s disease or as an adjuvant strategy to MCI treatment. Our hypothesis for this fact is that there is not enough pre-clinical experimental data that supports the clinical use of it. In this way, our group is working on many projects to add information that can give this support. We add this comment to the text (lines 52 to 59).

II. Methods.

1. Treatments. The authors must justify the dose applied and present the calculations. Unfortunately, the reference they provide does not have it.

Answer: Lithium, in the form of lithium carbonate (Li2CO3), was dissolved in water and administered ad libitum. The dose used was the same as published by our research team before for mice treatment (Nunes et al., 2015; Malerba et al., 2021) and proportional to that used in a previous manuscript with Alzheimer’s disease patientes (Nunes et al., 2013). The dose used in mice corresponds to 1.5 mg of lithium carbonate/day or 0.006 mEq of lithium/Kg. The dose was adjusted according to the mice's pharmacokinetics profile (Rouaud et al., 1993; Wood et al., 1986). The final dose was 5ug/day/animal, corresponding to 0.25 mg/kg/day. The treatment lasted for a total of 8 months (from 2 to 10 months of age). The authors added this information to the text (lines 107-113).

2. Experimental design. The authors use only females in their study. Please clarify and justify it. 

Answer: The rationale for exclusively utilizing female subjects in our study is explicated on lines 442 to 453. In this investigation, we opted to focus solely on female mice due to the well-documented higher prevalence of Alzheimer's disease in women compared to men. This observation has been supported by numerous studies, including those conducted by Niu et al. (2017), Beam et al. (2018), MacCarthy and Raval (2020), Meng et al. (2023). The prevailing hypothesis attributes this gender-specific discrepancy to the longer life expectancy of women compared to men. This argument and the references were added to the text (Lines 442-453).

3. Behavioral tests. Barnes maze. The authors apply a very unusual protocol for the Barnes maze, which generally assesses spatial memory acquisition and recall. And here. The authors repeated the task when animals were 5, 8, and 10 months old to assess memory recovery. So, please define the term "memory recovery" and present a reference showing the method.

Answer: We appreciate the reviewer's attention to the Barnes maze protocol employed in our study. While we understand the concern regarding the characterization of our protocol as "unusual," we would like to emphasize that our research group has previously utilized a similar Barnes maze protocol in other published works. For further details and reference, please consult the following articles, which are also cited in the text:

Nunes MA, Schöwe NM, Monteiro-Silva KC, Baraldi-Tornisielo T, Souza SI, Balthazar J, Albuquerque MS, Caetano AL, Viel TA, Buck HS. Chronic Microdose Lithium Treatment Prevented Memory Loss and Neurohistopathological Changes in a Transgenic Mouse Model of Alzheimer's Disease. PLoS One. 2015 Nov 25;10(11):e0142267. doi: 10.1371/journal.pone.0142267. 

Malerba HN, Pereira AAR, Pierrobon MF, Abrao GS, Toricelli M, Akamine EH, Buck HS, Viel TA. Combined Neuroprotective Strategies Blocked Neurodegeneration and Improved Brain Function in Senescence-Accelerated Mice. Front Aging Neurosci. 2021 Aug 23;13:681498. doi: 10.3389/fnagi.2021.681498.

As for the term "memory recovery," in the context of our study, it pertains to the evaluation of spatial memory at different time points during the aging process (5, 8, and 10 months old). The repeated administration of the Barnes maze task allows us to assess the potential recovery or decline in spatial memory performance over time, providing insights into age-related changes in memory function. 

4. Evaluation of the density of senile plaques with Thioflavin S. The authors used a solution of thioflavin S (Sigma T1892) to stain amyloid plaques in SAMP-8 mice. The same method has been used by numerous labs in the transgenic AD animals. However, aged SAMP-8 mice, in contrast to AD mice, do not present amyloid-beta plaques detectable with thioflavin S (7). Please explain or apply another staining (IHC) for amyloid to show changes. 

Answer: We appreciate the Reviewer's diligence in examining our methodology for the evaluation of senile plaques with Thioflavin S. As correctly pointed out, the use of Thioflavin S to stain amyloid plaques is a well-established method commonly applied in transgenic AD animal models. While it is noted that aged SAMP-8 mice, in contrast to AD mice, were previously reported not to present amyloid-beta plaques detectable with Thioflavin S (Reference 7, Pouquet et al., 2015), we would like to draw attention to more recent findings. In a study conducted by our group and others, including Malerba et al. (2021) and Guo et al. (2022), Thioflavin S was able to stain senile plaques, as evidenced by the observed fluorescence in brain tissue from SAMP-8 animals. These newer investigations, including our own, suggest that Thioflavin S was capable of staining senile plaques in SAMP-8 mice. We have incorporated this latest reference into the manuscript (line 489-490) to reinforce the validity of our results and the chosen methodology.

III. Results.

1. Fig. 2. The authors applied repeated measures ANOVA to show the effects of treatment with low-dose Lithium on spatial memory, however, the presentation of the results is not correct. You cannot demonstrate statistical significance with repeated measures ANOVA by the way you did it (at the points of measurements) instead, you must analyze the entire plot and compare the curves. 

Answer: We would like to clarify that there might be a misunderstanding regarding the presentation of results. Contrary to the perception that we analyzed the points of measurements individually, we indeed conducted a comprehensive analysis of the entire plot, comparing the curves across the treatment groups. Furthermore, to ensure the robustness of our findings, we employed the Bonferroni post-test to verify the differences among specific points. It's important to note that this approach aligns with our established and validated method for analyzing repeated measures ANOVA, which has been consistently applied in our previous works. For reference, we invite the reviewer to consult the following publications where similar methodologies were successfully employed:For references, please verify: Baraldi et al., Experimental Gerontology, 2013; Nunes et al., Plos One, 2015; Morzelle et al., Plos One, 2016; Malerba et al., Frontiers in Aging Neuroscience, 2021). We hope this clarification addresses the concerns raised.

2. Fig. 8. You are only displaying artifacts, no plaques. Look at reference #7 fig. 3. 

Answer: We appreciate the Reviewer's attention to Figure 8 and the concern raised regarding the presence of amyloid plaques. We acknowledge the reference provided (#7, Fig. 3) and understand the importance of accurate representation in our figures. We would like to clarify that, in our study, we took meticulous steps to ensure the reliability of our findings. Specifically, we conducted a thorough analysis of the negative control to eliminate any potential artifacts. However, we understand that the Reviewer has highlighted the need for a clearer demonstration of amyloid plaques in Figure 8. In both this manuscript and a previously published work, as referenced earlier, we have consistently demonstrated that SAMP-8 animals exhibit a higher prevalence of amyloid plaques compared to SAMR-1 animals of the same age. The clarity of our images supports the indication of the presence of these plaques.

IV. Discussion. There is no molecular mechanism explaining the effects of low-dose Lithium.

I propose a theory that could be a good fit.

1. In 1986 Adlercreutz et al. showed that Lithium lowers renal, cardiac and splenic ornithine decarboxylase activity (ODC) in mice (8).

2. In 1992 Gad M. Gilad Gad Gilad et al. demonstrated that chronic lithium treatment suppresses ornithine decarboxylase (ODC) activity in the brain (9).

3. Kan et al. (2015) convincingly demonstrated that pharmacologic disruption of the arginine utilization pathway by an inhibitor (DMFO) of arginase and ornithine decarboxylase protects the mice from AD-like pathology (10).

4. Polis et al. (2018) showed that arginase inhibition reverses cognitive decline and synaptic loss in a murine model of Alzheimer's Disease (11).

Accordingly, to enhance the quality of their work and deliver a unique and noteworthy research paper, the authors must employ additional techniques to investigate the hypothesis based on the findings of various groups about the influence of lithium salts on arginase and ODC expression levels and activities. It might be WB, IHC, or another method. I believe you have the lysates and the brains to run an experiment. 

Answer: The proposed theory linking Lithium to the regulation of ODC activity and its implications in Alzheimer's Disease pathology is intriguing. The references provided, particularly Adlercreutz et al. (1986), Gilad et al. (1992), Kan et al. (2015), present compelling evidence on the connection between Lithium, ODC activity, and AD-like pathology. We acknowledge the importance of investigating these pathways further to enhance the depth of our understanding. While we find the proposed hypothesis to be of great interest and relevance, it's important to note that the primary focus of the present work was to examine the effects of low-dose Lithium on the density of proteins specifically related to memory formation and modulation. Regrettably, at this juncture, we do not have sufficient lysates to undertake additional experiments exploring the suggested molecular mechanisms. However, we genuinely appreciate the Reviewer's constructive suggestion and acknowledge the potential for future investigations in this direction. We will duly consider pursuing these avenues in subsequent projects where we can allocate the necessary resources to delve deeper into the molecular mechanisms involved. Your feedback is valuable, and we look forward to exploring these connections in future research endeavors.

V. It is necessary to improve the quality of scientific English language. This includes correcting any errors in spelling, grammar, and punctuation to make the text clearer and more precise. 

Answer: Thank you for the suggestion. A review of the English writing was done.

Reviewer #2: Comments to Authors

In the present manuscript, "Multiple mechanisms of microdose lithium protect behavioral deficits and molecular mechanisms for memory formation in SAMP-8, a mouse model of accelerated aging", authors examined the effects of microdose lithium on behavioral abnormalities, anxiety, plaque formation and neuroplasticity in SAMP-8 mice. The study is planned and executed well and findings are interesting and relevant to the current research in the area of Alzheimer's disease. However, the following minor changes are required for the further improvement of the manuscript.

Title

1. The title is vague kindly rewrite it to make it clearer. 

Answer: This title was suggested by other Reviewers. We can change it again to: "Microdose lithium improves behavioral deficits and modulates molecular mechanisms of memory formation in female SAMP-8, a mouse model of accelerated aging."

2. The word "mechanism" is used twice in the title, please avoid repetition.

Answer: The authors thank you for the suggestion. The title was changed.

Abstract

1. Alzheimer's disease (AD) is the most common affection.

Affection does not seem a suitable word. 

Answer: The authors changed the word “affection” to “neuronal disorder” (Line 2). 

Introduction

• Kindly state the objective/s of the study at the end of the introduction section.

Answer: The objective of the work was stated in lines 57-59. 

Methods

Evaluation of neurons and astrocyte densities in the hippocampus: Page 10

1. Images of the hippocampal regions: GD, CA1, and CA3 were acquired using the Leica DMi8 177 inverted microscope.

What is meant by GD here? Is dentate gyrus (DG)? 

Answer: Yes. The complete information (granular layer of the dentate gyrus - GrDG) was added to the text (line 206). 

Results

1. Figure 2A, B, and C: Provide the figure legend to represent the groups such as SAMR-1 and SAMP-8 

Answer: As Figures A, B, and C have the groups’ representation in D, E, and F, respectively, we put the legend in these three graphs. The information was reinforced in the figure’s legend to become more clear (lines 284-286). 

2. Figure 2: Compare the results of SAMP-8 Li+ with SAMR-1 to further clarify the effects of lithium microdose. 

Answer: A comparison was done in the description of Figure 2 (lines 274-276).

3. Figure 4D: What could be the possible reason for the decrease of CaMK4 following low-dose lithium treatment? Please explain. 

Answer: There is no decrease in CAMKIV following treatment with lithium. What is shown in Figure 4D is a decrease in CAMKIV in SAMP-8 animals that was not prevented with lithium treatment. 

In the process of memory formation, Camk-IV plays an important role in activating the CREB transcription factor, promoting cell proliferation, differentiation, and protein synthesis (Benarroch, 2 Neurology 91(3): 125-132, 2018).

As stated, the difference shown in fig. 4D is between SAMR-1 animals and SAMP-8 animals without treatment. SAMR-1 animals do not show a cognitive decline. Therefore, the higher concentration of CAMK-IV in these animals may mean higher phosphorylation of CREB and greater maintenance of memory consolidation mechanisms, unlike what happens in SAMP-8 animals. In this way, the treatment with low-dose lithium did not influence the expression and density of this protein. This conclusion was added to the text (lines 335 – 336).

Reviewer #3: This paper addresses the effects of microdose lithium carbonate treatment (2-10 months of age) on Alzheimer-related brain phenotypes in the SAMP-8 mouse model of aging. Cognition was assessed using the Barnes maze longitudinally. Elevated plus maze was used to assess anxiety related behaviors. After treatment, brains were collected and stained for neurons, astrocytes, Thio-s and lysates were used to probe synaptic proteins and GSK-3Bser9 phosphorylation.

While the study is important and well written, there are major issues that call into question the reliability of the data presented.

1. One major issue is that it is impossible to know how much Lithium each individual mouse received, as these may vary across the cage. 5 ug/day/animal. Is the entire mixture consumed daily? If ad libitum, this suggests the dose fluctuates. 

Answer: The authors agree with the Reviewer that there is no way to ensure the amount of lithium each animal receives. However, as the treatment was taken for 8 months uninterruptedly, we can assure that the blood concentration was stable and effective as per our experience in previous reports from our research team in both mice and human beings (Nunes et al., 2013; Nunes et al., 2015; Malerba et al, 2021). Unfortunately, we did not measure the blood concentration of lithium in this group, but we made the same protocol with male SAMP-8 mice, measured their lithemia, and found a stable concentration (these data were submitted recently for publication). 

2. More details on SAMR-1 and SAMP-8 mice are needed. What are these mice? How are they generated?

Answer: The accelerated aging mouse line comes from AKR/J animals. In 1968, at Kyoto University, it was noticed, after brother-sister crossings, that some litters showed characteristics, such as decreased activity, periophthalmic lesions, and early death, but without any evidence of malformation. This new phenotype presented was inherited by other generations (TAKEDA, Neurobiology of aging 20(2): 105-110, 1999).

In 1975, the litters that presented the phenotypes described above were isolated as "senescence-prone" parents, and the animals that did not have an altered life course were "senescence-resistant" parents (TAKEDA, Neurobiology of aging 20(2): 105-110, 1999).

Today, it is known that there are several lineages of SAMP animals; the isolation of these lineages occurred due to the different characteristics presented between them. SAMP-8 animals present the hallmarks of accelerated aging, early death, memory and learning deficits, presence of amyloid plaques in the hippocampus, blood-brain barrier dysfunction, loss of neurons, and cortical atrophy (MIYAMOTO et al., Physiology & behavior 38(3): 399-406, 1986; YAGI et al., Journal of Neuropathology & Experimental Neurology 48(5): 577-590, 1989).

This information was added to the Methods (lines 74-85).

3. Line 89. There should be a SAMR-1 Lithium treated group. 

Answer: The use of SAMR-1 animals in this study was to control SAMP-8 animals. The differences found between the two groups show that molecular mechanisms are altered in animals with accelerated aging. Our group understands that the focus of the study was to evaluate the effects of lithium microdoses in SAMP-8 animals, SAMR-1 animals only prove that SAMP-8 animals present spontaneous molecular and behavioral changes.

4. Line 152, 153, etc: typo in phosphorylated

Answer: Thank you for the observation. The corrections were made. 

5. Line 164: typo

Answer: Thank you for the observation. The correction was made

6. Line 183: Concern that this method is not reliable. The detection range should not be altered between slices.

Answer: We thank the referee for the observation and we agree. There must be some mistake. The paragraph was rewritten (lines 213-214). 

7. Line 184: relations should be relation

Answer: The sentence was dismissed.

8. Line 192: typo

Answer: Thank you for the observation. The correction was made.

9. Line 204: Overall there are issues with the statistics. 2way ANOVA described in methods (with repeated measures). Looks like in some places only a t-test is used. The methods describe a 2way ANOVA with molecular assays, but there are not enough levels in the SAMR-1, because there was no lithium treated group, so it is unclear what the two-way ANOVA was comparing. P values of 0.05 are listed in the methods but not used if tests used were actually multiple t-tests. 

Answer: There was a typo error. The test used in molecular data was one-way ANOVA.

10. Fig 2A, 2D: It is unclear why after day 2, latency to enter escape box and number of errors are worse than baseline for the SAMP-8 mice.

Answer: There is no statistical difference on day 2 for any group. This means that the behavior on day 2 is similar to day 1. Only from day 3 do animals begin to express their behavioral differences in the maze. 

11. Figure 2C: If the SAMP-8 mice were divided to receive Li treatment, why do the SAMP-8 Li+ group already look better? Should have checked that the groups were balanced/equal before continuing with treatment. 

Answer: Animals were randomly assigned to the groups, and there were no statistical differences between the groups at that point.

12. Fig. 3: All individual data points should be shown.

Answer: Figures 5 to 8 were changed and now all individual points are shown.

13. Line 283: typo

Answer: Thank you for the observation. The correction was made

14. Line 278: Full uncropped blots should be provided in supplement. Additionally, a single lane from one of each mouse is not sufficient. 

Answer: We appreciate the reviewer's request for full uncropped blots in the supplementary materials and understand the importance of providing comprehensive data. In our experimental approach, due to the optimization of experiments and the limited availability of cellular lysates, we employed a strategy where the same membrane was incubated with different proteins of varying molecular weights. To ensure the integrity and reliability of our results, we performed Ponceau staining on the membranes, confirming the successful transfer of all proteins and the efficacy of electrophoresis. This quality control step was essential to validate the accuracy of our Western blot data. In response to the Reviewer's suggestion, we have included in the supplementary materials the complete membranes stained with Ponceau, allowing for a comprehensive view of the entire lanes. Additionally, we have provided membranes with all the individual lanes for transparency and thoroughness. These supplementary data aim to enhance the transparency and reproducibility of our Western blot analyses.

15. Figure 4: It is unclear how these differences are tested. It should be one way ANOVA with appropriate post-hoc test. Appear to be multiple t-tests.

Answer: We thank the Reviewer for the observation. There was a mistake in the description of statiscal methods that was corrected (line 239). The test used was indeed one-way ANOVA. 

16. Line 326: abbreviation "a" appears as "b" in Fig 4H.

Answer: The change was made.

17. Line 328: n of 6 in 3C and F but legend says n of 7.

Answer: We recognize these details and aim to offer clarification. In specific groups, the sample size is indeed 6 subjects, a decision made to exclude outliers during the analysis. This meticulous step was taken to enhance the robustness and accuracy of our results. Your attention to detail is invaluable, and we are grateful for the opportunity to address and rectify this discrepancy in our manuscript. If you have any further questions or concerns, please do not hesitate to reach out.

18. Overall, the use of WB for quantitation of synaptic proteins is antiquated because of its low sensitivity. Immunofluorescent staining and confocal imaging of serial sections with appropriate co-stains is more accurate.

Answer: We appreciate the reviewer's observation and acknowledge the consideration of alternative techniques, such as immunofluorescent staining and confocal imaging, for the quantitation of synaptic proteins. While we recognize that WB may have lower sensitivity compared to the mentioned methods, it is a well-established technique for evaluating protein expression. Although it may not offer the same level of accuracy as immunofluorescent staining and confocal imaging of serial sections, WB provides a valuable insight into the expression profile of these proteins. For future research endeavors, we are grateful for the suggestion to explore alternative methodologies and will duly consider incorporating immunofluorescent staining and confocal imaging with appropriate co-stains. We understand the importance of employing the most precise and advanced techniques available to enhance the accuracy of our findings. Thank you for your insightful feedback, and we look forward to incorporating these considerations into our future work.

19. It appears that the Actin band is re-used in Figs 4A, B, E, F and G. While it might be possible to probe multiple proteins of different molecular weights on the same blot, use secondaries with separate fluorescent channels, or even strip the plot to reprobe, the shape of the bands should be similar. In this case, Fig 4B is concerning because while the Actin bands are catching upward, the AMPA bands are catching downwards, suggesting they are from different gels/blots. The Actin and AMPA bands must be from the same blot. Showing the full blot would clear this up.

Answer: As explained above, the original images were included in the supplemental material.

20. Fig. 5,6,7: Neuron number cannot be quantified by NeuN percent area. Use stereology.

Answer: Thank you for your suggestion. We agree that neuron quantification should be done by stereology. Our data show neuron semi-quantification based on the proportional labeled area of NeuN, which could be associated with changes in neuronal density.

21. Fig. 8: Should be labeled thio-s not AB. Thio-S can stain many other structures beyond just AB. Moreover, the Thio-s staining is not convincing.

Answer: We thank the Reviewer and we explained above the rationale of using Thio-S in this work to show senile plaques.

22. Figure 8J: Individual data points from 5 mice should be shown.

Answer: All graphs were changed and now it is possible to visualize the individual data points.

23. Line 405: typo.

Answer: Thank you. The correction was made.

24. Line 406: this quote should be removed. This idea has since been dismissed. The increased prevalence for AD in women is equal to men at ages where the death rate is equal. Mice, which do not undergo menopause, do however show a lifeline decline in estrogen sensitivity in their tissues and this has been shown to relate to exacerbated pathology.

Answer: The authors respectfully disagree with the Reviewer's suggestion to remove the quote from Lines 446-447, as recent studies continue to provide evidence regarding the differences in the prevalence of AD between men and women. Contrary to the notion that this idea has been dismissed, recent research, including the review by Subramaniapillai et al. (2021) and the article by Dubal (2020), supports the existence of disparities in AD prevalence based on gender. These studies emphasize the ongoing relevance of considering gender-specific factors in AD pathology. While it is acknowledged that mice do not undergo menopause, it's important to note that they exhibit a decline in estrogen sensitivity in their tissues, which has been demonstrated to relate to exacerbated pathology (Frick, 2009; Koeble et al., 2016). Therefore, the inclusion of the quote is intended to reflect the existing discourse on this topic and the evolving understanding of gender-related factors in AD. If there are additional studies or perspectives that the Reviewer recommends considering, we would be open to discussing them and potentially incorporating relevant information into the manuscript. Thank you for your thoughtful review, and we are committed to ensuring the accuracy and up-to-date nature of our literature references. A comment with references was included in the Discussion (lines 442 – 453).

Frick KM. Estrogens and age-related memory decline in rodents: what have we learned and where do we go from here? Horm Behav. 2009 Jan;55(1):2-23. Doi: 10.1016/j.yhbeh.2008.08.015. Epub 2008 Sep 16. PMID: 18835561; PMCID: PMC2664384.

Koebele SV, Bimonte-Nelson HA. Modeling menopause: The utility of rodents in translational behavioral endocrinology research. Maturitas. 2016 May;87:5-17. doi: 10.1016/j.maturitas.2016.01.015. Epub 2016 Feb 3. PMID: 27013283; PMCID: PMC4829404.

Dubal DB (2020) Sex difference in Alzheimer's disease: An updated, balanced and emerging perspective on differing vulnerabilities. Handbook of clinical neurology 175: 261-273.

Subramaniapillai S, Almey A, Rajah MN, Einstein G (2021) Sex and gender differences in cognitive and brain reserve: Implications for Alzheimer's disease in women. Frontiers in Neuroendocrinology 60: 100879.

---

## [Decision Letter · Decision Letter 1]

13 Feb 2024

Microdose lithium improves behavioral deficits and modulates molecular mechanisms of memory formation in female SAMP-8, a mouse model of accelerated aging

PONE-D-23-36699R1

Dear Dr. Viel,

We’re pleased to inform you that your manuscript has been judged scientifically suitable for publication and will be formally accepted for publication once it meets all outstanding technical requirements.

Kind regards,

Kyung-Wan Baek, Ph.D.

Academic Editor

PLOS ONE

Additional Editor Comments (optional):

The manuscript has been enhanced to meet the standards required for publication in PLOS ONE. Please follow the instructions provided by the editorial office and proceed with the necessary procedures.

Reviewers' comments:

Reviewer's Responses to Questions

**Comments to the Author**

1. If the authors have adequately addressed your comments raised in a previous round of review and you feel that this manuscript is now acceptable for publication, you may indicate that here to bypass the “Comments to the Author” section, enter your conflict of interest statement in the “Confidential to Editor” section, and submit your "Accept" recommendation.

Reviewer #1: All comments have been addressed

Reviewer #2: All comments have been addressed

Reviewer #3: All comments have been addressed

2. Is the manuscript technically sound, and do the data support the conclusions?

Reviewer #1: Yes

Reviewer #2: Yes

Reviewer #3: Yes

3. Has the statistical analysis been performed appropriately and rigorously? 

Reviewer #1: Yes

Reviewer #2: Yes

Reviewer #3: Yes

4. Have the authors made all data underlying the findings in their manuscript fully available?

Reviewer #1: Yes

Reviewer #2: Yes

Reviewer #3: Yes

5. Is the manuscript presented in an intelligible fashion and written in standard English?

Reviewer #1: (No Response)

Reviewer #2: Yes

Reviewer #3: Yes

6. Review Comments to the Author

Reviewer #1: The article has undergone significant improvements, making it suitable for publication in Plos One journal. The language has been refined, and the data has been presented in a clear and concise manner.

Reviewer #2: All the comments are properly addressed by the authors and I have no concerns regarding the manuscript and recommend the acceptance of the article.

Reviewer #3: (No Response)

7. PLOS authors have the option to publish the peer review history of their article (what does this mean?). If published, this will include your full peer review and any attached files.

Reviewer #1: **Yes: **Baruh Polis

Reviewer #2: **Yes: **Sanila Amber

Reviewer #3: No

---

## [Editor Report · Acceptance letter]

22 Mar 2024

PONE-D-23-36699R1 

PLOS ONE

Dear Dr. Viel, 

I'm pleased to inform you that your manuscript has been deemed suitable for publication in PLOS ONE. Congratulations! Your manuscript is now being handed over to our production team.

Kind regards, 

on behalf of

Dr. Kyung-Wan Baek 

Academic Editor

PLOS ONE